# A strong constraint on radiative forcing of well-mixed greenhouse gases

Jing Feng[1✉], David Paynter[2], Raymond Menzel[2] & Ryan Kramer[2]

Radiative forcing from well-mixed greenhouse gases (WMGHGs) is a main driver of Earth's energy imbalance and global surface climate change[1,2]. It remains difficult to constrain, largely because its longwave (LW) instantaneous radiative forcing (IRF) component depends on atmospheric state and is subject to radiative parameterization error[3–7]. The IRF measures the immediate change in radiative fluxes at the tropopause[8–10] caused by perturbations in WMGHG concentrations. Here we show that increasing WMGHG concentrations have enhanced LW IRF by $3.69 \pm 0.07$ W m$^{-2}$ (95% confidence interval) since 1850. We first use global line-by-line radiative transfer simulations to provide a global benchmark of LW IRF for the main WMGHGs under realistic, all-sky conditions. We then identify a robust linear relationship between LW IRF and outgoing longwave radiation (OLR), enabling state-dependent LW IRF to be directly inferred from regressions against satellite-observed OLR. Furthermore, LW IRF explains 91% of the inter-model spread in effective radiative forcing (ERF, which includes rapid atmospheric adjustments beyond the IRF) for $CO_2$ (ref. 11) across Earth system models. Benchmarking model-simulated IRF using the regression technique reveals that most discrepancies originate from radiation parameterizations and correcting LW IRF biases would reduce uncertainty in $CO_2$ ERF by 50%. Our results establish a simple and robust framework for quantifying state-dependent radiative forcing of WMGHGs, providing an observation-informed pathway for future climate assessments.

Earth's radiative energy balance is fundamentally regulated by greenhouse gases (GHGs), which absorb and re-emit infrared radiation, exerting an influence on the planet's climate system. Anthropogenic emissions of well-mixed greenhouse gases (WMGHGs), including $CO_2$, $CH_4$, $N_2O$, chlorofluorocarbons (CFCs) and hydrofluorocarbons (HFCs), perturb this balance by reducing outgoing longwave radiation (OLR), thereby inducing global warming until a new equilibrium is achieved. The magnitude of this perturbation is characterized by instantaneous radiative forcing (IRF), defined as an instantaneous change in net radiative fluxes at the top of the atmosphere (TOA) or tropopause[8–10].

Recent decades have brought notable advances in our ability to constrain the radiative forcing of WMGHGs. Long-term gas concentration records from observational networks such as the National Oceanic and Atmospheric Administration (NOAA) Earth System Research Laboratories (ESRL) and NASA Advanced Global Atmospheric Gases Experiment (AGAGE)[12,13], combined with improvements in molecular spectroscopy and line-by-line radiative transfer modelling[14], have enabled high-fidelity benchmark estimates of IRF for selected atmospheric profiles[15,16]. Under carefully sampled conditions, line-by-line calculations have provided high-confidence estimates of IRF for clear-sky (cloud-free) conditions. For example, the most recent benchmark for the clear-sky longwave (LW) IRF owing to WMGHG increases from 1850 to 2014 is 2.66–2.69 W m$^{-2}$ at TOA and 3.63–3.66 W m$^{-2}$ at tropopause, with sampling errors of less than 0.015 W m$^{-2}$ relative to the 2014 climatology[16] (Extended Data Table 1). The benchmark has not been updated

in a decade, despite substantial increases in WMGHG concentrations since 2014 (ref. 2).

Despite these advances, benchmark estimates of radiative forcing remain limited to fixed clear-sky background conditions and do not capture the broader uncertainties associated with evolving surface temperatures, stratospheric states, water vapour and cloud distributions for $CO_2$ (refs. 5,7,17–19) and other WMGHGs[4,20–22]. More than 90% of WMGHG IRF arises from the LW spectrum[15,16], in which radiative fluxes are highly sensitive to background atmospheric conditions[23,24]. Line-by-line radiative transfer calculations using a limited set of observed cloud profiles can estimate all-sky IRF[15] but sampling errors become more uncertain owing to the spatio-temporal variability of clouds[25]. Moreover, when used as inputs to line-by-line calculations, persistent biases and uncertainties in humidity and cloud fields in reanalyses[26] and Earth system models (ESMs)[24,27] introduce further uncertainties that are challenging to constrain.

In climate assessment reports[1], uncertainty in radiative forcing is recognized as an important contributor to uncertainty in projected $CO_2$-induced warming[6]. Radiative forcing is inherently a model-diagnosed quantity and its uncertainty is commonly expressed through the spread of effective radiative forcing (ERF) across ESMs[28]. This spread is about 12%, with an even larger spread (about 15%) in the LW IRF component[7,18]. Yet it remains unclear whether these discrepancies arise from the model-simulated atmospheric fields or from the radiation parameterizations[3,5,7,16,17]. Although line-by-line calculations could, in principle,

[1]Atmospheric and Oceanic Sciences Program, Princeton University, Princeton, NJ, USA. [2]Geophysical Fluid Dynamics Laboratory, Princeton, NJ, USA. ✉e-mail: jing.feng@princeton.edu

provide benchmarks to isolate these contributions, their computational cost has so far precluded such comprehensive evaluations.

Independent of model discrepancies, satellite observations provide direct evidence of GHG forcing[29,30], but these signals are mixed with variability in surface temperature, water vapour, clouds, aerosols and short-lived climate forcers. Isolating the GHG component requires radiative transfer simulations based on retrieved or assumed background conditions[29,31], introducing uncertainties often exceeding 10%, comparable with the inter-model spread in ESMs.

Together, these limitations underscore a central challenge: despite advances in spectroscopy, satellite observations and radiative modelling, the state dependence of LW IRF remains a leading source of uncertainty in the radiative forcing of GHGs.

## Global-scale benchmark of LW IRF

Enabled by the GPU-optimized line-by-line radiative transfer code (GRTcode) developed at NOAA's Geophysical Fluid Dynamics Laboratory, this study presents the first, to our knowledge, global-scale, decadal line-by-line calculations of LW radiative forcing for $CO_2$, $CH_4$, $N_2O$, CFCs and HFCs. The code is benchmarked in ref. 16. These calculations use monthly mean surface temperature, air temperature, water vapour, ozone and cloud from the ERA5 reanalysis dataset[32]. Annual mean WMGHG concentrations follow the Coupled Model Intercomparison Project Phase 7 (CMIP7) input datasets from 2001 to 2022 (ref. 13), with $CO_2$, $CH_4$ and $N_2O$ extended to 2024 using NOAA ESRL trends[33], as detailed in Methods. Following ref. 12, radiative effects of ozone-depleting substances are combined as CFC-12 equivalence (CFC12-eq) and other fluorinated gases as HFC-134a equivalence (HFC134a-eq). Results generated following this experiment set-up are referred to as GRTcode–ERA5.

The GRTcode–ERA5 dataset provides gridded, monthly mean OLR from 2001 to 2024 for a 'control' experiment. To isolate the LW IRF of WMGHG $x$, relative to pre-industrial (denoted as 'PI', 1850) concentrations, sensitivity experiments ($x$PI) are conducted that are identical to the control experiment, except that gas $x$ is held fixed at its pre-industrial concentration. Details are provided in Methods. For each gas $x$, the LW IRF at vertical level $i$ is computed as the difference in net downward LW flux between the control and perturbation ($x$PI) experiments:

$$F_{i,x} = (R_{\downarrow,i,x\mathrm{PI}} - R_{\uparrow,i,x\mathrm{PI}}) - (R_{\downarrow,i,\mathrm{control}} - R_{\uparrow,i,\mathrm{control}}), \tag{1}$$

in which $R_{\uparrow,i}$ and $R_{\downarrow,i}$ denote the upwelling and downwelling, respectively, LW radiative fluxes at level $i$. Although the IRF at TOA is often used to decompose the TOA energy budget, the tropopause value is more relevant for surface temperature responses and adjusted forcing[8,10] and has been used as the key metric for radiative forcing since the first Intergovernmental Panel on Climate Change (IPCC) report[9]. In this paper, LW IRF at TOA is referred to as $\mathrm{IRF_{TOA}}$ and 'LW IRF' specifically refers to $F_i$ at the 200 hPa level[3], which serves as a consistent tropopause level for benchmark evaluations.

From 2001 to 2024, GRTcode–ERA5 simulations show an increase in LW IRF from 2.66 to 3.70 W m$^{-2}$ and at TOA from 1.85 to 2.49 W m$^{-2}$ (Fig. 1a). The increase in $CO_2$ from 370.8 to 423.0 parts per million by volume (ppmv) accounts for 0.863 W m$^{-2}$ of the increase, whereas $CH_4$, $N_2O$, HFC134a-eq and CFC12-eq contribute 0.067, 0.067, 0.048 and −0.012 W m$^{-2}$, respectively.

Compared with the recent benchmark in ref. 16, which estimated $\mathrm{LW\ IRF_{TOA}}$ from 100 ERA5 profiles in 2014, our global clear-sky LW IRF (3.68 W m$^{-2}$) and $\mathrm{LW\ IRF_{TOA}}$ (2.70 W m$^{-2}$) show excellent agreement (Extended Data Table 1). Pincus et al.[16] did not provide a benchmark for all-sky conditions but estimated $\mathrm{LW\ IRF_{TOA}}$ between 1.98 and 2.07 W m$^{-2}$, based on the all-sky to clear-sky IRF ratio from three CMIP6 models[31]. Our GRTcode–ERA5-derived all-sky $\mathrm{LW\ IRF_{TOA}}$ for 2014 is 2.16 W m$^{-2}$,

exceeding the upper bound of this range, suggesting weaker LW cloud effects in ERA5 than the three CMIP6 models included in ref. 16.

Although ERA5 probably provides more realistic atmospheric conditions by assimilating all-sky satellite observations[32], potentially improving persistent biases in earlier reanalysis datasets[26], simulating cloud radiative effects in line-by-line models remains sensitive to assumptions about subcolumn cloud inhomogeneity[25], vertical cloud overlap[34] and hydrometeor particle sizes[35–37]. These sensitivities lead to biases and uncertainties that are challenging to quantify.

## Constraining present-day radiative forcing

Despite potential uncertainties in all-sky forcing estimates, the LW IRF of different WMGHGs exhibits highly consistent spatial patterns (Fig. 1b; see also Extended Data Fig. 1), largely independent of cloud conditions. This consistency aligns with the inhomogeneous forcing structure previously identified for $CO_2$ (ref. 5). We find that the spatial variability in IRF closely follows local OLR under both clear-sky and all-sky conditions (Extended Data Figs. 1 and 2). This strong correspondence suggests that temperature, humidity and clouds influence OLR[23,24] and LW IRF in similar ways: IRF tends to be higher in regions with warmer surface temperatures[23] and lower where relative humidity or cloud amount is higher[24].

Motivated by this relationship, we construct a simple regression model for each WMGHG using GRTcode–ERA5 results under clear-sky conditions using year 2010 as the reference:

$$F = a(R_{2010\mathrm{WMGHG}} - b) - r \tag{2}$$

in which $F$ is the LW IRF and $R_{2010\mathrm{WMGHG}}$ is OLR with WMGHG concentrations at the 2010 reference level. The regression intercept $b$ is obtained from Extended Data Fig. 3 and the slope $a$ is then determined as $a = (\overline{R} - b)\overline{F}$, in which $\overline{R}$ and $\overline{F}$ denote the global mean clear-sky OLR and IRF in 2010 compared with 1850. Under this definition, the residual ($r$) is 0 by construction for clear-sky conditions.

To extend the model beyond the 2010–1850 forcing range, we perturb each WMGHG concentration up to four times its 2010 level (Methods) and compute the regression slopes for every forcing scenario. These perturbations allow us to describe $a$ as a polynomial function of WMGHG concentration (Tables 1 and 2). IRF induced by individual WMGHGs is additive, with a small global mean residual of 0.01 W m$^{-2}$ (about 0.5%).

When applying this regression model to time-varying WMGHG concentrations for 2001 to 2024, we directly used the OLR computed from the control (Methods) GRTcode–ERA5 experiment for each year (neglecting the small differences from $R_{2010\mathrm{WMGHG}}$). In clear-sky conditions, the regression approach accurately reproduces the spatial pattern of clear-sky LW IRF for all gasses (Extended Data Fig. 2, 24-year average) and achieves global mean accuracy comparable with five benchmark line-by-line models[16] (Extended Data Table 1).

Notably, this linear relationship remains the same under all-sky conditions. Using all-sky OLR in equation (2), the clear-sky-derived regression model accurately predicts the all-sky LW IRF–OLR relationship (grey line in Fig. 1c; $R^2 = 0.99$), the spatial distribution of LW IRF (Fig. 1d) and the zonal mean profiles for each gas (Fig. 1e; dashed versus solid curves). The global mean bias ($r$ in equation (2)) under all-sky conditions remains within 0.01 W m$^{-2}$ for present-day forcing (Table 1). Further evaluations of individual gasses and the $\mathrm{IRF_{TOA}}$ results are provided in Methods and Extended Data Figs. 3 and 5. Conditions for this linear relationship to hold are further discussed in Methods and Extended Data Fig. 4. These statistics confirm that regions with similar OLR, regardless of atmospheric or cloud conditions, exhibit nearly identical LW IRF.

The regression estimates yield excellent agreement with directly computed LW IRF values (dashed lines versus circles in Fig. 1a). To assess uncertainty, we evaluate deviations in the regression estimates

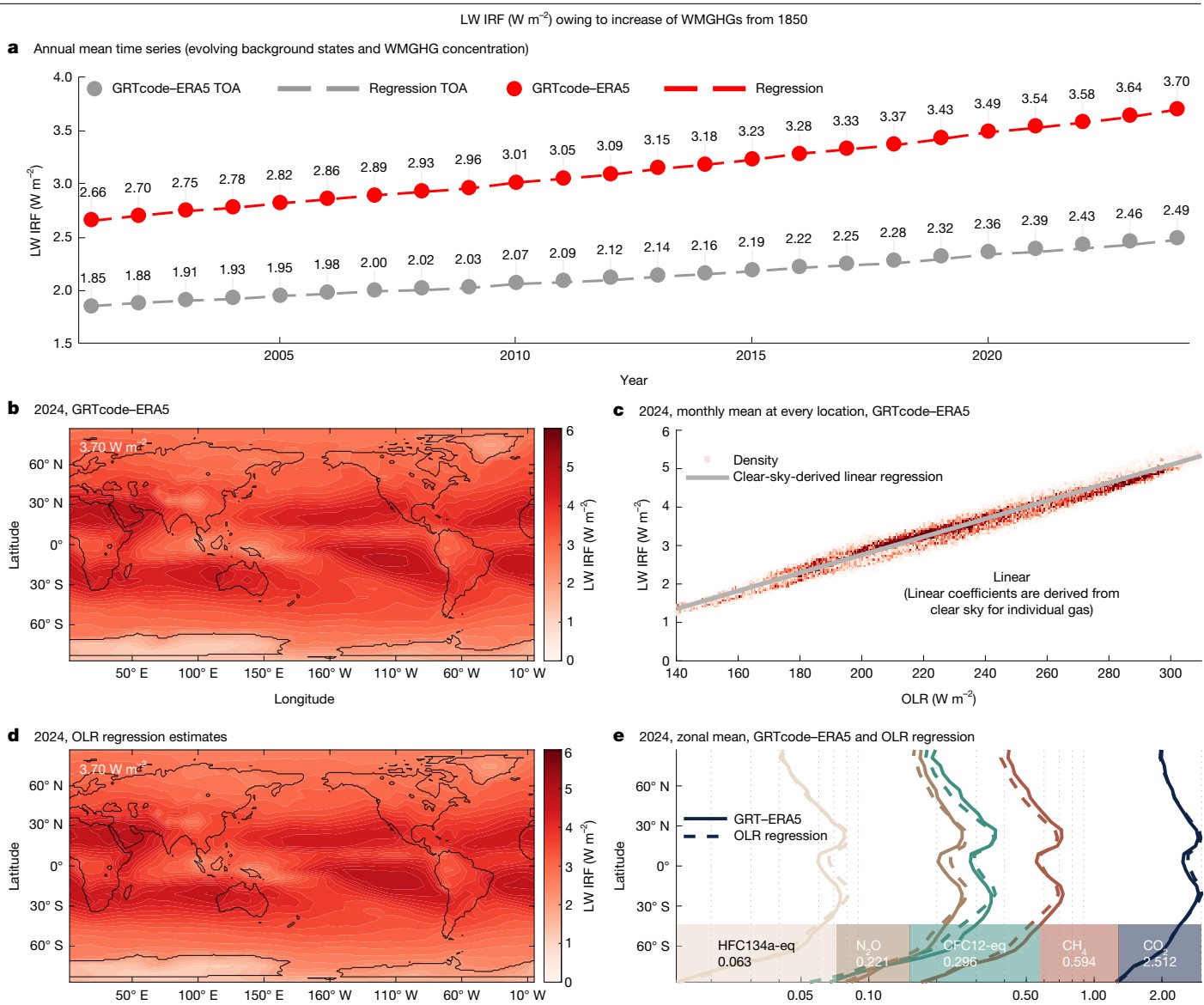

**Fig. 1 | Global line-by-line calculations reveal a strong linear relationship between LW IRF and OLR. a**, Global annual mean LW IRF (red, tropopause; grey, TOA) time series from GRTcode–ERA5[47] (circles), regression estimates based on evolving WMGHG concentrations[13,33] and OLR from GRTcode–ERA5 (dashed lines). **b–e**, Evaluation of LW IRF at the tropopause from WMGHG increases (1850–2024) under 2024 GRTcode–ERA5 background conditions: annual mean LW IRF from GRTcode–ERA5 (**b**); joint histogram of monthly LW IRF (*y*-axis) and OLR (*x*-axis) compared with regression estimates based on clear-sky relationships (grey line) (**c**); LW IRF reconstructed from spatial OLR using the regression model (**d**); and zonal mean LW IRF from GRTcode–ERA5 (solid lines) and regression applied to zonal mean OLR of GRTcode–ERA5 (dashed lines) for the main WMGHGs (**e**).

when applied to monthly and meridional means over 24 years (blue scatters, Extended Data Fig. 5). The corresponding OLR ranges from 215 to 267 W m⁻², spanning diverse atmospheric states and far exceeding the variability in global mean (black scatters, Extended Data Fig. 5). The uncertainty at the 95% confidence level, estimated as ±1.96 times the standard error, is 0.064 W m⁻² (about 2%). Although GRTcode–ERA5 may include biases in atmospheric or cloud fields, the resulting uncertainty remains representative as long as the observed global mean state of Earth lies within the wide sampled range. Thus, the OLR–regression method provides a conservative and observation-based estimate of LW IRF, independent of potential biases in GRTcode–ERA5.

We next apply this regression model (equation (2) and Table 1) to CERES EBAF v4.2 satellite observations of OLR[38] to obtain observationally constrained estimates of LW IRF relative to pre-industrial levels. The OLR in the CERES EBAF product uses an objective constrainment algorithm[39] that adjusts radiation fluxes for consistency with heat

storage in the Earth-atmosphere system; the remaining global mean OLR uncertainty is ±2.5 W m⁻² (ref. 38). This yields annual time series of LW IRF for individual WMGHGs over 2001–2024 (Fig. 2). Over this 24-year period, the CERES-constrained LW IRF for WMGHGs increases from 2.65 to 3.69 W m⁻². The uncertainty range (reported as the 95% confidence interval) is 0.073 W m⁻² ($\sqrt{0.064^2 + 0.015^2 + 0.032^2}$, including 0.064 W m⁻² from the regression method (Extended Data Fig. 5), 0.015 W m⁻² from clear-sky line-by-line calculations (Extended Data Table 1) and 0.032 W m⁻² from the uncertainty in OLR. A more detailed discussion of the uncertainty can be found in Methods and a breakdown of the uncertainty related to each WMGHG is presented in Extended Data Fig. 5.

In summary, the regression model derived from clear-sky GRTcode–ERA5 enables accurate evaluations of the LW IRF driven by time-varying WMGHG concentrations, using all-sky OLR to describe the atmospheric state. This provides a tight observational constraint on present-day

**Table 1 | Regression coefficients for LW IRF of WMGHGs at tropopause**

| Gas | $a$[a] | $b$ (W m$^{-2}$) | $r$ (W m$^{-2}$)[b] | Uncertainty (%) |
|---|---|---|---|---|
| $CO_2$ | $0.0358\ln(C/C_0) + 0.0015\ln(C/C_0)^2$ | 63 | 0.010 | 1.2 |
| $CH_4$ | $0.0089(\sqrt{C} - \sqrt{C_0}) - 0.0006(\sqrt{C} - \sqrt{C_0})^2$ | 110 | 0.003 | 4.0 |
| $N_2O$ | $0.0271(\sqrt{C} - \sqrt{C_0})$ | 100 | 0.005 | 3.3 |
| CFC12-eq | $2.712(C - C_0) - 45.44(C - C_0)^2$ | 124 | 0.017 | 6.6 |
| HFC134a-eq | $1.424(C - C_0) - 17.016(C - C_0)^2$ | 124 | 0.001 | 8.1 |
| Sum | | | −0.005 | |

The forcing follows the form $F = a(R - b) - r$, in which $R$ is the OLR with present-day WMGHG concentrations. The coefficient $a$ is a function of gas concentration (ppmv), in which $C_0$ is the unperturbed concentration at the 2010 level ($CO_2$: 388.901 ppmv; $CH_4$: 1.8097 ppmv; $N_2O$: 0.3232 ppmv; CFC12-eq: 0.0010 ppmv; HFC134a-eq: $2.06 \times 10^{-4}$ ppmv) and $C$ is perturbed concentrations validated for the range from PI to 4×present day. The term $r$ is the global mean bias in all-sky IRF for $C$ at pre-industrial concentration (0 W m$^{-2}$ in clear sky). These regression coefficients are derived from clear-sky monthly mean, gridded GRTcode–ERA5 (ref. 47) results at year 2010 and are evaluated using all-sky monthly mean meridional mean, providing the relative uncertainty (95% confidence interval) of the regression-based LW IRF. [a]The quadratic terms in $(C - C_0)$ make a small contribution. They are included to maintain relatively consistent accuracy across wide ranges. [b]Evaluated with respect to 2010–1850 forcing; biases ($r$) are dependent on concentration and can be evaluated using an open-source package[47].

LW IRF and, given its global mean accuracy comparable with line-by-line calculations (Extended Data Table 1), offers an efficient tool to benchmark radiative forcing under diverse atmospheric conditions in ESMs. In the next section, we apply this framework to evaluate the forcing of $4 \times CO_2$ simulated by ESMs.

## Implications to Earth system modelling

In ESMs, ERF with fixed sea surface temperature ($F_{fsst}$) is diagnosed as the TOA energy imbalance following rapid atmospheric adjustments to GHG perturbations. These values are commonly reported for $2 \times CO_2$ and $4 \times CO_2$ scenarios and serve as essential metrics for quantifying equilibrium climate sensitivity[1]. In IPCC AR6 (ref. 1), $F_{fsst}$ was reported as 3.71 ± 0.44 W m$^{-2}$ (95% confidence interval) for $2 \times CO_2$ and 7.98 ± 0.76 W m$^{-2}$ for $4 \times CO_2$, based on multi-model CMIP6 simulations[11].

In some ESMs, the IRF can be diagnosed using a second radiation call at every model time step, for which absorber concentrations are perturbed while all other model fields remain unchanged. The fluxes from this diagnostic-only 'double call' do not feed back on the model physics and represent the instantaneous response to the specified forcing agent. For WMGHGs, such double-call perturbations are available only for $4 \times CO_2$ in Atmospheric Model Intercomparison Project (AMIP) experiments and only for a limited subset of six ESMs. As a result, $4 \times CO_2$ is at present the only scenario for which IRF can be directly diagnosed in these models. Because ESMs must rely on computationally efficient radiation parameterizations rather than spectrally resolved line-by-line radiative transfer, such double-call diagnostics provide a critical means of evaluating how well the internal radiation scheme of each model captures the intended forcing response.

Using the double-call diagnostics from the six ESMs and corresponding ERF values from AMIP $4 \times CO_2$ simulations (Extended Data Table 4), we find that ERF (8.10 ± 1.12 W m$^{-2}$) strongly correlates with LW IRF (8.82 ± 1.31 W m$^{-2}$, $R^2 = 0.91$; Fig. 3a) but shows a much weaker correlation with IRF$_{TOA}$ ($R^2 = 0.48$; Extended Data Table 4). Thus the spread in LW IRF is a dominant cause of spread in ERF across these models. It confirms the relevance and predictive skill in using tropopause-defined forcing versus TOA for radiative forcing and climate change.

To trace the source of inter-model spread in LW IRF, we conduct offline radiative transfer simulations using monthly mean atmospheric conditions from each model, as described in Methods. These simulations are performed under clear-sky conditions using a fast RTE-RRTMGP parameterization code[40], which isolates radiative responses from the model-specific representations of cloud processes. A known +0.33 W m$^{-2}$ bias in RTE-RRTMGP has been removed following benchmark studies[16]. The resulting LW IRF values exhibit good agreement across models (black points, $x$-axis of Fig. 3b), with a spread of only 0.2 W m$^{-2}$, much smaller than the 1.31 W m$^{-2}$ spread seen in double-call diagnostics. Notably, this spread is not driven by clouds: the variability under clear-sky double-call diagnostics (blue) is nearly identical to that in all sky (yellow), indicating that discrepancies arise mainly from differences in radiation parameterizations.

Although line-by-line calculations under all-sky conditions could, in principle, benchmark LW IRF for each model, they are computationally prohibitive and cannot fully represent model-specific cloud–radiation interactions. Our regression framework provides an efficient alternative, yielding accurate LW IRF estimates across diverse atmospheric and cloud states characterized by the OLR of each model.

The same regression model (equation (2)) is applied to estimate the LW IRF induced by pre-industrial and quadrupled pre-industrial $CO_2$ concentrations as

$$F_{4\times CO_2} = 0.0512(R_{2010WMGHG} - 63 \text{ W m}^{-2}) - r, \tag{3}$$

in which the slope of 0.0512 is computed as the difference in the regression coefficient $a$ between pre-industrial ($a = -0.0110$) and quadrupled pre-industrial $CO_2$ ($a = 0.0402$) using Table 1. The global

**Table 2 | Regression coefficients for instantaneous LW radiative forcing of WMGHGs at TOA**

| Gas | $a$ | $b$ (W m$^{-2}$) | $r$ (W m$^{-2}$) | Uncertainty (%) |
|---|---|---|---|---|
| $CO_2$ | $0.0359\ln(C/C_0)$ | 155 | 0.007 | 5.1 |
| $CH_4$ | $0.0113(\sqrt{C} - \sqrt{C_0}) - 0.0008(\sqrt{C} - \sqrt{C_0})^2$ | 148 | −0.010 | 3.5 |
| $N_2O$ | $0.0369(\sqrt{C} - \sqrt{C_0}) - 0.0024(\sqrt{C} - \sqrt{C_0})^2$ | 148 | −0.003 | 3.1 |
| CFC12-eq | $4.28(C - C_0) - 93.42(C - C_0)^2$ | 148 | −0.001 | 4.8 |
| HFC134a-eq | $2.32(C - C_0) + 37.44(C - C_0)^2$ | 148 | 0.003 | 7.7 |
| Sum | | | −0.010 | |

Coefficients are defined as in Table 1.

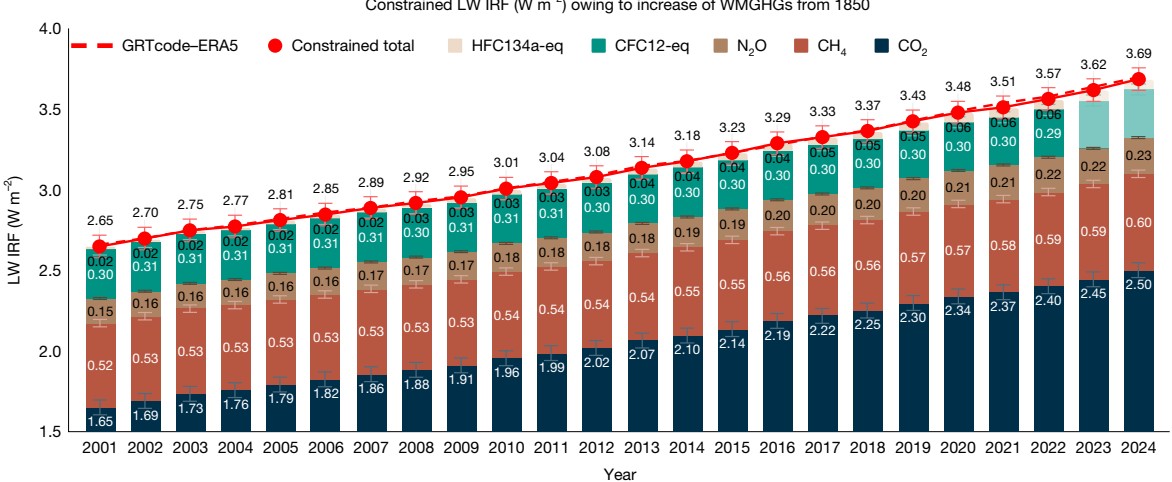

**Fig. 2 | Satellite-observed OLR tightly constrains present-day LW IRF of WMGHGs.** OLR-constrained LW IRF owing to increases in the main WMGHG concentrations relative to 1850 is estimated using clear-sky-derived linear coefficients from GRTcode–ERA5 (ref. 47), global mean time series of OLR from CERES EBAF v4.2 (ref. 48) and global mean time series of WMGHG concentrations. LW IRF from all-sky GRTcode–ERA5 is shown as a dashed red curve. Error bars denote the uncertainty range (95% confidence interval) of the constrained LW IRF, arising from regression uncertainty, gas spectroscopy and CERES radiative flux uncertainty (Methods).

mean bias is $r = 0.092$ W m$^{-2}$ and the uncertainty range is estimated to be ±0.10 W m$^{-2}$, based on 24-year monthly and meridional means as described in the previous section and Methods.

The regression method (equation (3)) reproduces the LW IRF very well, with the regression estimates (red curve) closely matching the GRTcode–ERA5 calculations (red markers) over the 24-year period. The LW IRF increases from 8.96 W m$^{-2}$ in 2001 to 9.08 W m$^{-2}$ in 2024. The reference OLR, $R_{2010WMGHG}$, is obtained by removing the LW IRF$_{TOA}$ relative to 2010 while following the same regression framework (equation (2) and Table 2), such that only differences caused by variations in temperature, humidity and clouds affect OLR. When applied to CERES OLR, the IRF and its increase over time is highly consistent with GRTcode–ERA5. This increase reflects record-high OLR associated with surface warming[2] and a weakened cloud LW effect. The ability of the regression method to accurately estimate

LW IRF under anomalous OLR conditions highlights its robustness in a changing climate.

Notably, the regression uncertainty (0.10 W m$^{-2}$) is smaller than the inter-model spread across five independent line-by-line models (0.19 W m$^{-2}$; Extended Data Table 1). Therefore, the regression-based IRF estimates, when driven by OLR simulated by each ESM, serve as an efficient alternative to line-by-line calculations for benchmarking IRF under each model's climatology. Before analysing the double-call diagnostics produced by each model's radiation scheme, we evaluate the regression method in its ability to address model discrepancies. Figure 3b compares the clear-sky IRF computed offline using model-simulated climatology with those estimated by the regression method, shown as black markers. The agreement is excellent, with deviations well within the regression uncertainty. Similar comparisons for 24 more experiments are presented in Extended Data Fig. 6. These

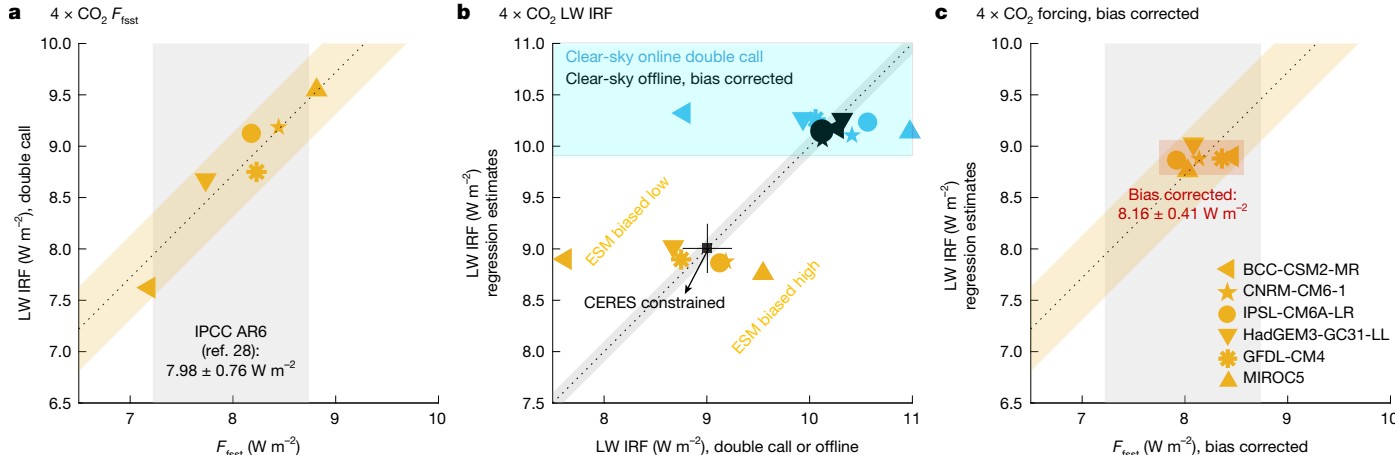

**Fig. 3 | LW IRF explains most discrepancies in 4 × CO$_2$ ERF ($F_{fsst}$) across ESMs.** **a**, Comparison of ERF ($x$-axis) with LW IRF ($y$-axis) from online double-call diagnostics[28,49]. **b**, Comparison of LW IRF from the OLR–regression method ($y$-axis) with online double-call diagnostics (yellow, all-sky; blue, clear-sky) and offline clear-sky RTE-RRTMGP including a 0.33 W m$^{-2}$ bias correction (black), using their respective OLR (data listed in Extended Data Table 4). The black square shows the CERES-constrained LW IRF (9.00 W m$^{-2}$, 2001–2024). The error bars indicate the total uncertainty (95% confidence interval, ±0.24 W m$^{-2}$)

arising from the regression method (±0.10 W m$^{-2}$, grey shading), CERES OLR (±0.09 W m$^{-2}$) and gas spectroscopy (±0.20 W m$^{-2}$; see Extended Data Fig. 5 and Methods). **c**, Repeat of **a** after applying model-specific IRF bias corrections from **b**. Yellow shading in **a** and **c** shows the 2.5–97.5% range of differences between ERF and LW IRF, grey shading indicates the ERF range in IPCC AR6 (7.98 ± 0.76 W m$^{-2}$), and red shading in **c** shows the bias-corrected ERF (8.16 ± 0.41 W m$^{-2}$).

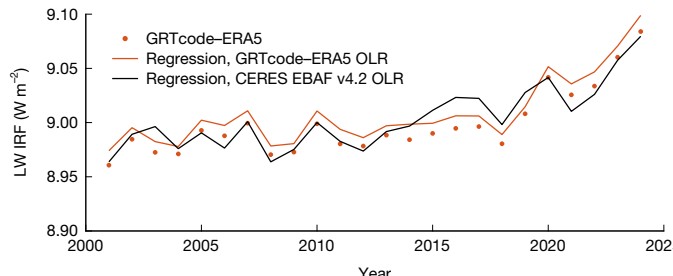

**Fig. 4 | LW IRF caused by quadrupling pre-industrial CO₂.** LW IRF of 4 × CO₂ (relative to pre-industrial) at evolving background states from 2001 to 2024 from GRTcode–ERA5 (ref. 47) (7.5° resolution; red markers), from the regression method applied to GRTcode–ERA5 OLR (red line) and from the regression method applied to CERES EBAF v4.2 OLR[48] (black line).

results confirm the accuracy of the regression method independently of the GRTcode–ERA5 evaluation conducted earlier.

Extending the regression to all-sky OLR derived from the ESMs' double-call diagnostics (Fig. 3b, *y*-axis), we find an all-sky LW IRF of $8.89 \pm 0.15$ W m$^{-2}$, consistent with the CERES-constrained range of $8.96$–$9.08 \pm 0.24$ W m$^{-2}$ (Fig. 4). The reduced inter-model spread suggests that ESMs simulate the observed global mean OLR well during the historical period. Deviations from the regression-predicted IRF are interpreted as model-specific radiation biases. Assuming that the reduced IRF spread translates to a corresponding reduction in ERF spread, we derive a bias-corrected fixed sea surface temperature forcing, $F_{fsst} = 8.16 \pm 0.41$ W m$^{-2}$, for 4 × CO₂. This represents a substantial improvement over the original inter-model spread of $8.10 \pm 1.12$ W m$^{-2}$, reducing the spread by about 60%.

The remaining spread (yellow shading in Fig. 3c) arises from discrepancies in shortwave IRF ($-0.52 \pm 0.48$ W m$^{-2}$) and adjustment processes ($-0.20 \pm 0.38$ W m$^{-2}$). Unlike the LW component, a simple relationship between TOA fluxes and shortwave IRF is less likely to hold, as surface albedo primarily controls reflection, whereas clouds not only reflect but also mask portions of the shortwave IRF, effects that TOA fluxes alone may not fully distinguish. Alternative formulations[41,42], along with improved TOA[43] and surface[44] shortwave observations, may help to better constrain these uncertainties. Because the adjustment process is not independent of radiation[7,21,45], further reductions in inter-model spread could be achieved if consistent radiation parameterizations were used.

## Conclusion

This study establishes the first global benchmark for the LW IRF of WMGHGs under realistic, all-sky conditions. Using a spectrally resolved, highly parallelized line-by-line radiative transfer model, we quantify a post-1850 LW IRF (at tropopause) of $3.69 \pm 0.07$ W m$^{-2}$ (95% confidence interval) by 2024, with 38% of this increase occurring since 2001. Confidence in this benchmark is reinforced by a regression method that integrates observational constraints from satellite-observed OLR to account for uncertainties from cloud effects and evolving climate states. In the future, such tight constraints for the LW IRF can only be achieved through the continuation of stable, long-term observational records of energy fluxes[46].

For climate projection simulations, we demonstrate that LW IRF at the tropopause is a dominant source of inter-model spread in CO₂-induced ERF. The demonstrated skill of the OLR–regression method provides a practical alternative to computationally intensive line-by-line diagnostics for benchmarking LW IRF. When applied to ESMs, this approach isolates and corrects model-specific biases in radiation schemes, reducing the inter-model spread in CO₂-induced ERF by approximately 50% relative to the spread assessed in the IPCC AR6 (ref. 1).

By linking physically robust line-by-line calculations with observational constraints, this study provides a scalable pathway to reducing the persistent uncertainties in GHG forcing[6]. Community-wide use of the OLR–regression method, together with double-call diagnostics applied systematically to each main WMGHG, would enable consistent benchmarking of radiative transfer parameterizations in ESMs, strengthening confidence in climate assessments and long-term climate projections.

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

## Methods

### Line-by-line calculations

The Geophysical Fluid Dynamics Laboratory's GPU-compatible radiation code (GRTcode) is a line-by-line radiative transfer code using HITRAN2016 (ref. 50) and MT_CKD 2.5 (ref. 51). It has been benchmarked in ref. 16, showing excellent agreement with other line-by-line radiative transfer models, including LBLRTM v12.8 (ref. 52) provided by Atmospheric and Environmental Research, RFM[53] calculation conducted by the Geophysical Fluid Dynamics Laboratory, 4AOP[54] provided by the Laboratoire de Météorologie Dynamique and ARTS 2.3 (ref. 55) provided by the University of Hamburg. We also note that although GRTcode does not include $CO_2$ line mixing, it agrees well with models (such as RFM and LBLRTM) that include the mechanism, which is consistent with the understanding that the line mixing induces negligible (within 1%) effects on the LW radiative forcing[14]. Details of this comparison are listed in Extended Data Table 1. Inter-model discrepancies are estimated as the standard deviations and are multiplied by 1.96 to infer 95% confidence interval in IRF caused by gas spectroscopy.

For clouds, the GRTcode uses a high-spectral-resolution cloud optics parameterization developed in ref. 37, which has been validated to achieve reasonable radiative closure with hyperspectral-resolution remote sensing instruments in the mid-infrared[56,57]. The cloud optics parameterization[37] is built on Mie-scattering calculations for liquid clouds and an optics library for severely roughened solid columns developed in ref. 58. On the basis of grid-cell-mean cloud fraction and cloud water content, subgrid cloud water content is stochastically generated at every wavenumber following ref. 25 and cloud optical depth is assumed to be uniform within each $1 cm^{-1}$ bin.

The all-sky calculations of GRTcode, with broadband cloud optics parameterizations, have shown good agreement in long-term global mean trend with spaceborne hyperspectral infrared sounders[59]. Sensitivity experiments have been conducted, using different configurations for the stochastic cloud generation and effective radius, and alternatively building the cloud optics parameterization on a state-of-the-art, irregularly shaped Voronoi ice model[60,61]. Although the maximum deviations could exceed 10% in the simulated all-sky IRF, these deviations do not affect the regression coefficients, which are derived from clear-sky conditions, nor the all-sky residual ($r$; Tables 1 and 2). The insensitivity of the IRF–OLR linearity to cloud parameterizations is consistent with the idealized simulations shown in Extended Data Fig. 4a–d and supports the robustness of the observationally constrained IRF presented in this study.

Using monthly mean surface temperature, humidity, ozone concentration and cloud conditions from the ERA5 (refs. 32,62) reanalysis dataset, a set of global-scale line-by-line experiments perturbing the concentrations of WMGHGs was conducted for the period 2001–2024 (ref. 47). These experiments include:
- Control: a control experiment using a time series of global mean, annual mean concentrations of WMGHGs[13].
- $x$PI: Identical to control except that an individual WMGHG species, $x$, is held fixed at its 1850 concentration ($CO_2$: 284.297 ppmv; $CH_4$: 0.7988 ppmv; $N_2O$: 0.2716 ppmv; $8.2 \times 10^{-6}$ ppmv for CFC12-eq and $2.02 \times 10^{-5}$ ppmv HFC134a-eq).
- $x2 \times$PI, $x3 \times$PI, $x4 \times$PI, $x4\times$: similar to $x$PI except that $x$ is two to four times the 1850 concentration (PI) or four times its annual mean concentration.

To minimize computational costs, we conduct radiative transfer calculations at a resolution of 2.5° for the year 2010, which is used to construct the regression model, and at a coarser resolution of 7.5° for other years that are used to evaluate the model. The resolution differences cause small (<0.5%) differences in the IRF. These experiments generate TOA OLR (W m$^{-2}$). For the period from 2001 to 2022, the time series of WMGHG concentrations is based on the CMIP7 greenhouse input dataset[13]. For the years 2023 and 2024, global mean concentrations of $CO_2$, $CH_4$ and $N_2O$ are taken from NOAA's Global Monitoring Laboratory[33,63], with a bias correction applied to ensure continuity with ref. 13. Concentrations of CFC12-eq and HFC134a-eq gases after 2022 are held fixed at their 2022 levels. Radiative transfer calculations are conducted at $0.1 cm^{-1}$ resolution; we have performed tests to demonstrate that increasing the resolution does not alter the broadband IRF. Extended Data Table 1 lists the clear-sky LW IRF estimated from GRTcode–ERA5 calculations for the year 2014. The difference between the GRTcode results submitted to ref. 16 is small and arises from differences in gas concentrations that have been updated since CMIP6.

### The linear OLR–regression method

**Construction.** Using the GRTcode–ERA5 simulation for the year 2010 and a first-order linear regression model (equation (2)), we derive coefficients ($a$ and $b$) from clear-sky simulations for both LW IRF and LW IRF$_{TOA}$ (Extended Data Fig. 3). Specifically, $a$ is derived from a range of forcing induced by changes in gas concentration, ranging from the 2010 level to one, two, three and four times the pre-industrial concentration and four times the 2010 concentration. This allows us to describe $a$ as a polynomial function of WMGHG concentrations (logarithmic dependence for $CO_2$ (ref. 64), square-root dependence for $CH_4$ and $N_2O$ (ref. 15) and linear dependence for CFCs and HFCs[65]), as summarized in Tables 1 and 2. The second-order terms listed in Tables 1 and 2 are included to preserve similar accuracy across a broad concentration range. Over the pre-industrial to 4 × present-day concentrations used in this study, the quadratic contribution is small and the LW IRF dependence for CFCs and HFCs (on the order of $10^{-3}$–$10^{-4}$ ppmv) remains effectively linear.

The global mean bias under all-sky conditions is derived as the residual when applying these coefficients with the all-sky OLR in the year 2010. Biases in IRF induced by WMGHG increase from 1850 to 2010 is listed in Tables 1 and 2. Biases for other perturbation scenarios are provided in the form of a look-up table and an example script in ref. 47.

**Explanation.** The clear-sky-derived linear regression model explains more than 90% of the monthly gridded variability in all-sky LW IRF and LW IRF$_{TOA}$. We find that this robust linear relationship holds for each WMGHG, despite their distinct absorption spectra, and it originates from dry atmospheric conditions without water vapour or clouds (yellow markers in Extended Data Fig. 7).

Idealized line-by-line experiments indicate that the linear OLR–IRF relationship arises from basic radiative transfer physics. Using the ERA5 zonal mean temperature profile with an isothermal stratosphere and a vertically uniform greybody absorber, IRF$_{TOA}$ is nearly linear with OLR. This linearity largely persists when random overlapping absorbers of varying optical depths and altitudes are added (Extended Data Fig. 4a–d) and breaks down only when the spectral width of the overlapping absorber varies with height (Extended Data Fig. 4e), the gas is not well mixed (Extended Data Fig. 4f) or strong temperature inversions occur (Extended Data Fig. 4g). These experiments demonstrate that a near-linear LW IRF–OLR relationship is a generic consequence of additive monochromatic radiative fluxes and thus expected for all WMGHGs under Earth-like conditions.

When overlapping with clouds, realistic absorption spectra (for example, $CO_2$ and $CH_4$) preserve the OLR–IRF relationship even more effectively than a grey absorber (Extended Data Fig. 5b,c). GRTcode–ERA5 simulations further show that clouds have negligible influence on the relationship for every main WMGHG (Extended Data Figs. 3 and 7), with or without water vapour. The small deviations observed in realistic atmospheres primarily arise from water vapour spectral overlap, which slightly alters the regression slope (Extended Data Fig. 7).

In the GRTcode–ERA5 results, the only exception to this linear relationship is the $CO_2$ IRF$_{TOA}$. This contrasts with the enhanced linearity found in the idealized experiment (Extended Data Fig. 4c) that assumed an isothermal stratosphere. The deviation arises because LW IRF$_{TOA}$

is sensitive to stratospheric temperature and lapse rate[7,19,66], both of which exhibit strong seasonal variability driven by the Brewer–Dobson circulation[67]. By contrast, OLR is insensitive to temperatures above the tropopause, as these layers only contribute to TOA in the strong absorption bands of $CO_2$ and $O_3$ and are partly masked by stratospheric $CO_2$ (ref. 68). It is well established that Earth's OLR is primarily determined by surface temperature[23], tropospheric relative humidity and clouds[24].

The LW IRF used in this study, defined at the tropopause, is largely insensitive to tropopause or stratospheric temperatures, similar to OLR. For $CO_2$, the LW IRF from upwelling fluxes at the tropopause shares the same sensitivity to tropopause temperature as the downwelling component[19], effectively cancelling the dependence on tropopause temperature. For other WMGHGs, whose stratospheric absorption is weak, defining IRF at the TOA or tropopause produces small differences. Because the downwelling fluxes at the tropopause are mainly controlled by the local temperature, at which WMGHGs are most abundant, the LW IRF is largely unaffected by stratospheric temperature or by stratospheric adjustment processes. Assuming that radiative forcing does not alter the stratospheric dynamical heating rate[8], the adjusted forcing is expected to converge to the same value at both the TOA and the tropopause. Given that the tropopause-level IRF is insensitive to stratospheric temperature, it more closely reflects the adjusted forcing and thus serves as a more robust indicator of surface climate change[1,3,8–10,69–71].

**Evaluation and uncertainties.** Applying the regression model to 24 years of OLR from GRTcode–ERA5 accurately reproduces the spatial pattern of LW IRF for each WMGHG (Extended Data Figs. 1 and 2). Even for $CO_2$-induced LW $IRF_{TOA}$, for which deviations from linearity are largest, the regression model explains 97% of the annual mean gridded variability, as the seasonal variations in stratospheric temperature have been removed. This model outperforms more complex analytical formulations that rely on meteorological conditions as inputs[18,19,72].

Uncertainties (reported as 95% confidence intervals) associated with the LW IRF derived from the OLR–regression method, $\sigma$, arise from three independent sources:
- Regression uncertainty ($\sigma_r$), quantified from deviations of the monthly, meridional mean all-sky LW IRF estimated by the regression method relative to GRTcode–ERA5 calculations over the 24-year period (blue scatters in Extended Data Fig. 5);
- Spectroscopic uncertainty ($\sigma_s$), represented by discrepancies among five independent line-by-line models for clear-sky radiative transfer (Extended Data Table 1); and
- Observational uncertainty ($\sigma_o$), estimated from the 2.5 W m$^{-2}$ uncertainty in the global mean OLR product and converted to LW IRF using the regression slope $a$.

The regression uncertainty $\sigma_r$ generally falls within 2% of the LW IRF, which is comparable with the spectroscopic uncertainty $\sigma_s$. These three independent sources are combined in quadrature as follows:

$$\sigma = \sqrt{\sigma_r^2 + a^2\sigma_o^2 + \sigma_s^2}. \qquad (4)$$

The individual contributions of each source of uncertainty for all WMGHGs are listed in Extended Data Fig. 5.

When the OLR–regression method is applied to other datasets, $\sigma_r$ and $\sigma_s$ remain fixed because they depend on the accuracy of the regression model and gas spectroscopy. The observational uncertainty $\sigma_o$, however, should be recalculated on the basis of the OLR product used. Also, the long-term instrumental stability of the OLR record is not explicitly considered here; when such stability evaluations become available, further drift in the IRF associated with OLR drift can be incorporated by scaling it through the regression slope $a$. Nevertheless, because $a$ is small (that is, 0.0512 W m$^{-2}$ for 4 × $CO_2$), the drift in IRF is expected to be much smaller than the drift in the OLR record.

**Applications.** The regression method provides a practical and computationally efficient tool for estimating LW IRF from changes in WMGHG concentrations between two states ($C_1$ and $C_2$). For consistent and accurate application across different contexts, we recommend the following steps:
- For each WMGHG, the regression slopes corresponding to concentrations $C_1$ and $C_2$ can be obtained from Tables 1 and 2 (relative to the 2010 reference level). The IRF associated with the change from $C_1$ to $C_2$ is then estimated from the difference between these two slopes, which provides a more accurate representation of concentration dependence.
- The regression coefficients were validated for $C_1$ and $C_2$ between pre-industrial and 4 × 2010 levels, covering the historical to near-future range. Extrapolation beyond this range may introduce bias and more uncertainty.
- Because the observed or simulated OLR is not at the 2010 reference concentration level, the WMGHG-induced LW $IRF_{TOA}$ component should be removed when evaluating trends driven by varying background climatology (as in Fig. 4). It should also be removed when the LW $IRF_{TOA}$ is large. For example, when applying the method to pre-industrial ESM experiments, the LW IRF at tropopause can be estimated as

$$F = a(R - b - 2.07 \text{ W m}^{-2}), \qquad (5)$$

in which $R$ is the simulated OLR under pre-industrial WMGHGs and 2.07 W m$^{-2}$ represents the reduction in OLR induced by 2010 WMGHGs relative to pre-industrial (Fig. 1).

To enable direct use for both model evaluation and observational analysis, example scripts are publicly available in ref. 47.

## Comparisons with existing IRF estimates

Alternative estimates of all-sky LW IRF can be derived from simplified formulations[1,15,73] (Extended Data Table 2) for IRF induced by WMGHG increases from 1850 to 2014, using conversion factors from line-by-line calculations with sampled cloud profiles[45,74] (Extended Data Table 3). For CFCs and HFCs, IPCC AR6 (ref. 1) gives a total of 0.36 W m$^{-2}$, slightly higher than the GRTcode–ERA5 benchmark (0.34 W m$^{-2}$) but within the 0.04 W m$^{-2}$ spectroscopy uncertainty. For $CO_2$, $CH_4$ and $N_2O$, refs. 15,73 yield totals of 2.75 and 2.64 W m$^{-2}$, respectively, compared with 2.83 W m$^{-2}$ from GRTcode–ERA5. The underestimation could arise from the conversion factor and, most likely, from overestimated cloud effects that could result from sampling or cloud–radiation interaction process.

The robust linear relationship between OLR and LW IRF presented in this study provides a simple rule of thumb: accurate LW IRF estimates can be obtained from limited sampling if the weighted mean OLR of the sampled atmospheric and cloud states matches the observed global mean. This finding suggests that biases in existing simplified formulas probably arise from lower mean OLR in the sampled conditions rather than from the functional form of the formulas themselves. Limited sampling is particularly useful because it enables computationally efficient calculations across a wide range of concentrations compared with global-scale line-by-line calculation. The OLR–regression model (equation (2)) can therefore be applied as an extra constraint when sampling the observed climate[15,20] or used to rescale forcing estimates from the sampled mean OLR to the OLR of a target climate scenario. Incorporating this approach enables more reliable estimates for extreme WMGHG perturbations across diverse climate conditions.

## RTE-RRTMGP calculations with ESM inputs

RTE-RRTMGP[40] is a Fortran library for fast global-scale radiative transfer calculations. As described in ref. 24, radiative transfer simulations at every model grid are conducted with inputs of monthly mean surface

temperature, temperature and humidity profiles from standard CMIP6 outputs at 19 pressure levels, climatological ozone profile and present-day WMGHG concentrations except for $CO_2$. Simulations are conducted with 1850 $CO_2$ concentrations and $4 \times CO_2$ concentrations for 28 model runs conducted by 11 CMIP6 models for three experiments (piClim-control, piClim-$4 \times CO_2$ and AMIP[49]), providing level-by-level downwelling and upwelling LW radiative fluxes. These radiative fluxes are used to interpret the global mean clear-sky OLR and LW IRF at each vertical level.

The radiative biases of RTE-RRTMGP can be evaluated using RFMIP[16] by comparing with line-by-line simulations for the same set of atmospheric profiles. The clear-sky LW IRF of $4 \times CO_2$ simulated by RTE-RRTMGP is 10.536 W m$^{-2}$, 0.33 W m$^{-2}$ higher than that by GRTcode for the same set of atmospheric profiles in ref. 16. This bias is removed in Fig. 3b and Extended Data Fig. 6 when compared with IRF estimated by the OLR–regression method. Although the results for AMIP experiments are included in Fig. 3 in black, other experiments are presented in Extended Data Fig. 6, further confirming that the OLR–regression method can accurately estimate the clear-sky LW IRF within the proposed uncertainty range (0.1 W m$^{-2}$).

## Data availability

The monthly mean ERA5 (ref. 32) reanalysis dataset is publicly available[62]. CMIP6 and CMIP5 experiments are accessible at https://aims2.llnl.gov/search and summarized in the Methods. Coastlines are generated using the Matlab/R2021b Mapping Toolbox[75]. WMGHG concentrations are combined using the CMIP7 input dataset[13] for years 2001 to 2022 and NOAA's Global Monitoring Laboratory[33,63] for years 2023 and 2024; both datasets are publicly accessible. Radiative fluxes generated using GRTcode–ERA5 are available for all experiments listed in the Methods at https://doi.org/10.5281/zenodo.17458936 (ref. 47).

## Code availability

Scripts to generate key results are available at https://doi.org/10.24433/CO.0711190.v2 (ref. 76). Scripts to derive LW IRF from the OLR–regression method are available at https://doi.org/10.5281/zenodo.17458936 (ref. 47). The GFDL line-by-line radiative transfer code is publicly accessible at https://github.com/fengzydy/GRTCODE/releases/tag/2025.08.Submission.

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

**Acknowledgements** We thank H. Letu for providing the advanced Voronoi multispectral ice cloud scattering model, which was essential for quantifying the effects of overlapping cloud spectrum. We thank F. Roemer and the anonymous reviewers for their detailed and constructive review comments that have helped improve this manuscript, and V. Naik, C. Fan and A. Pouyaei for an internal review of this study. We acknowledge computation resources from the Geophysical Fluid Dynamics Laboratory (GFDL) and Princeton Cooperative Institute for Modeling the Earth System (CIMES) made available for this research. This report was prepared by J.F. under award from the National Oceanic and Atmospheric Administration, US Department of Commerce. The statements, findings, conclusions and recommendations are those of the author(s) and do not necessarily reflect the views of the National Oceanic and Atmospheric Administration or the US Department of Commerce.

**Author contributions** J.F. conceived the research project, conducted the experiments, and designed the regression framework. J.F. and D.P. designed the experiments. R.M. and J.F. developed the GRTcode. J.F. and R.K. contributed to data processing and analyses. J.F. drafted the manuscript and all authors contributed to the initial submission.

**Competing interests** The authors declare no competing interests.

**Additional information**
**Correspondence and requests for materials** should be addressed to Jing Feng.

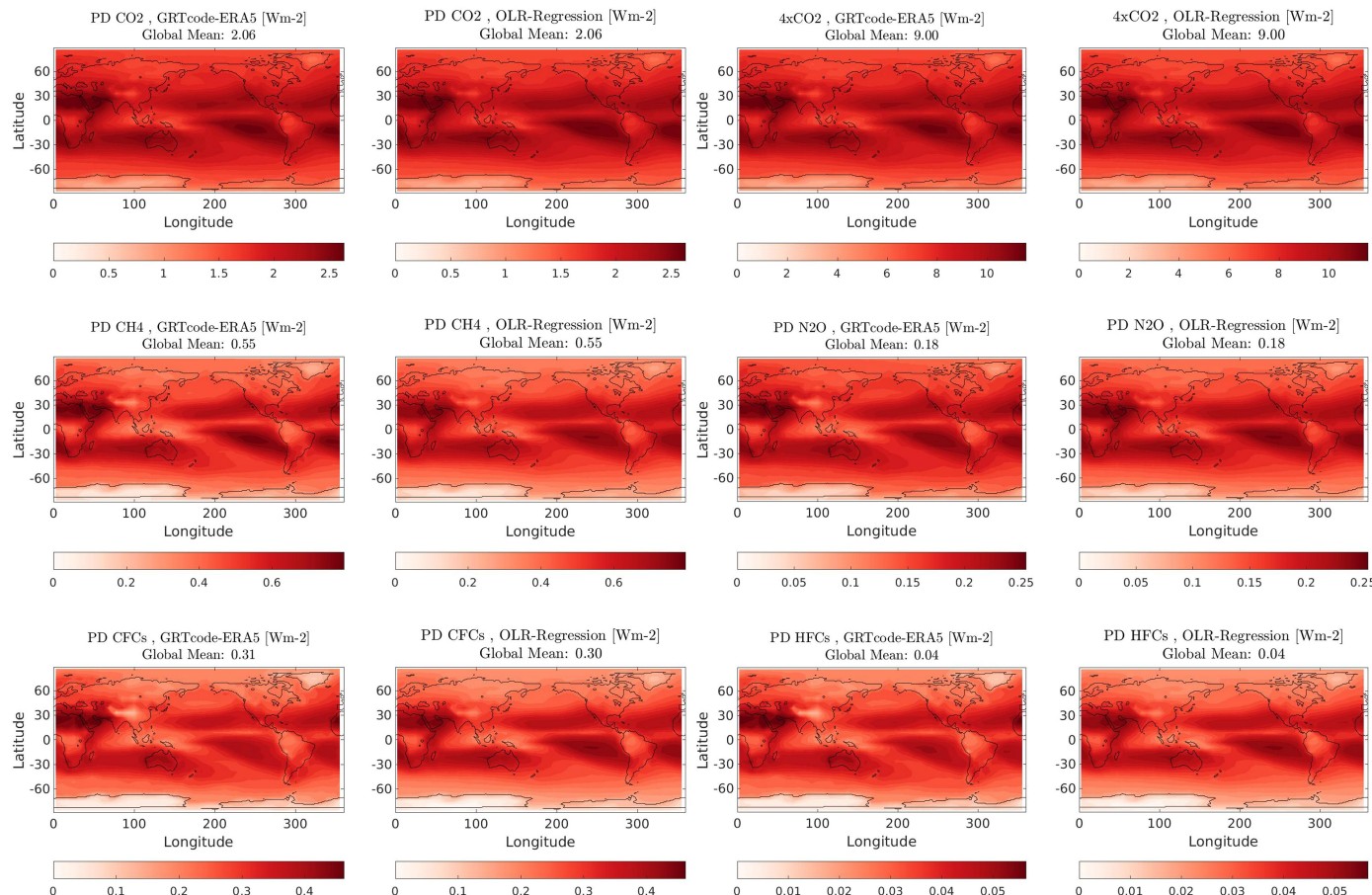

**Extended Data Fig. 1 | The OLR–regression method reproduces present-day (24-year mean, 2001–2024) LW IRF induced by the main WMGHGs from line-by-line calculations.** Change in present-day (PD) LW IRF relative to the year 1850 caused by $CO_2$, CH4, $N_2O$, CFCs, HFCs and $4 \times CO_2$, from GRTcode–ERA5 and the regression estimates based on OLR simulated by GRTcode–ERA5 (ref. 47).

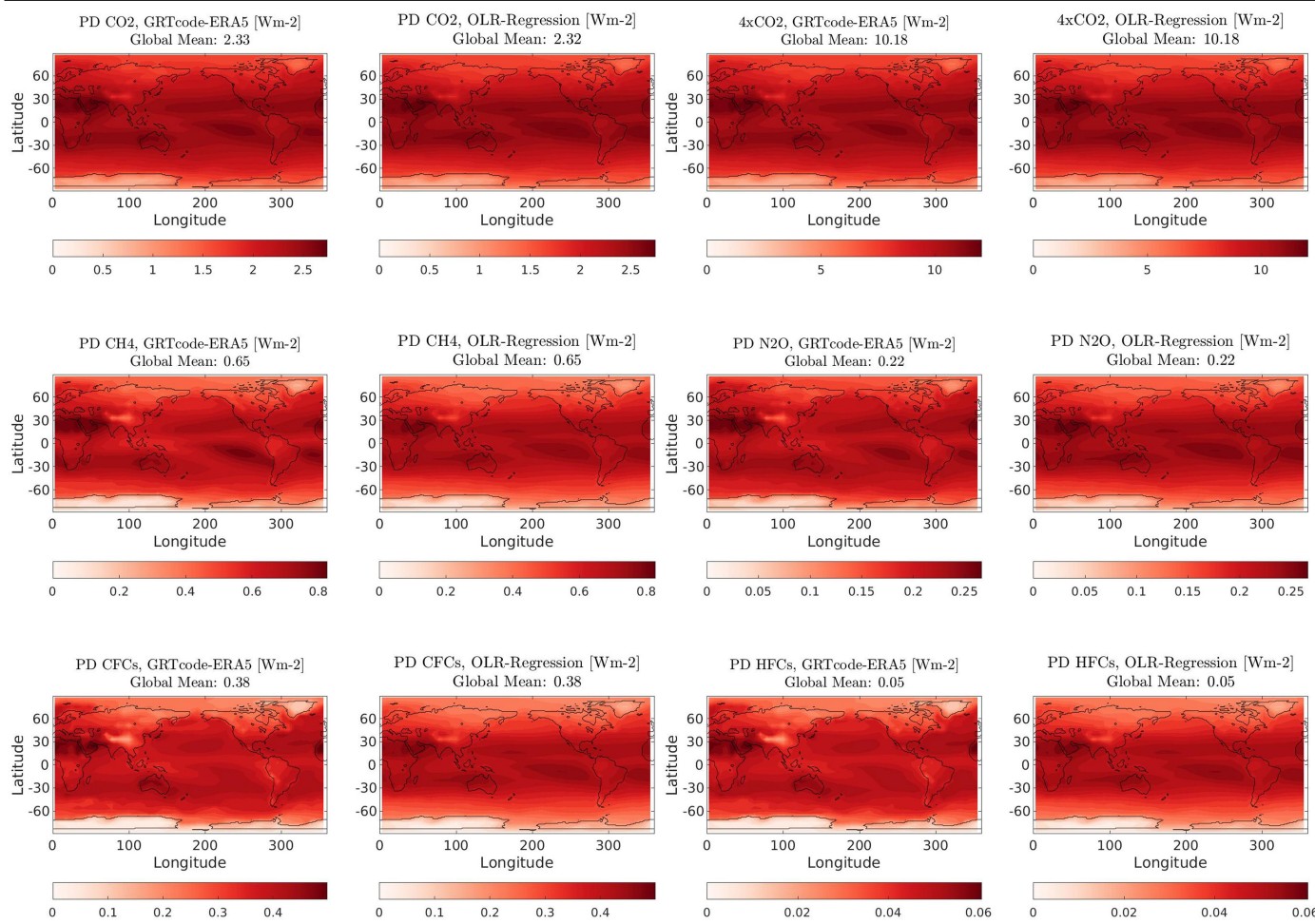

**Extended Data Fig. 2 | The OLR–regression method reproduces present-day clear-sky LW IRF from line-by-line calculations.** Similar to Extended Data Fig. 1 but for clear sky. PD, present day.

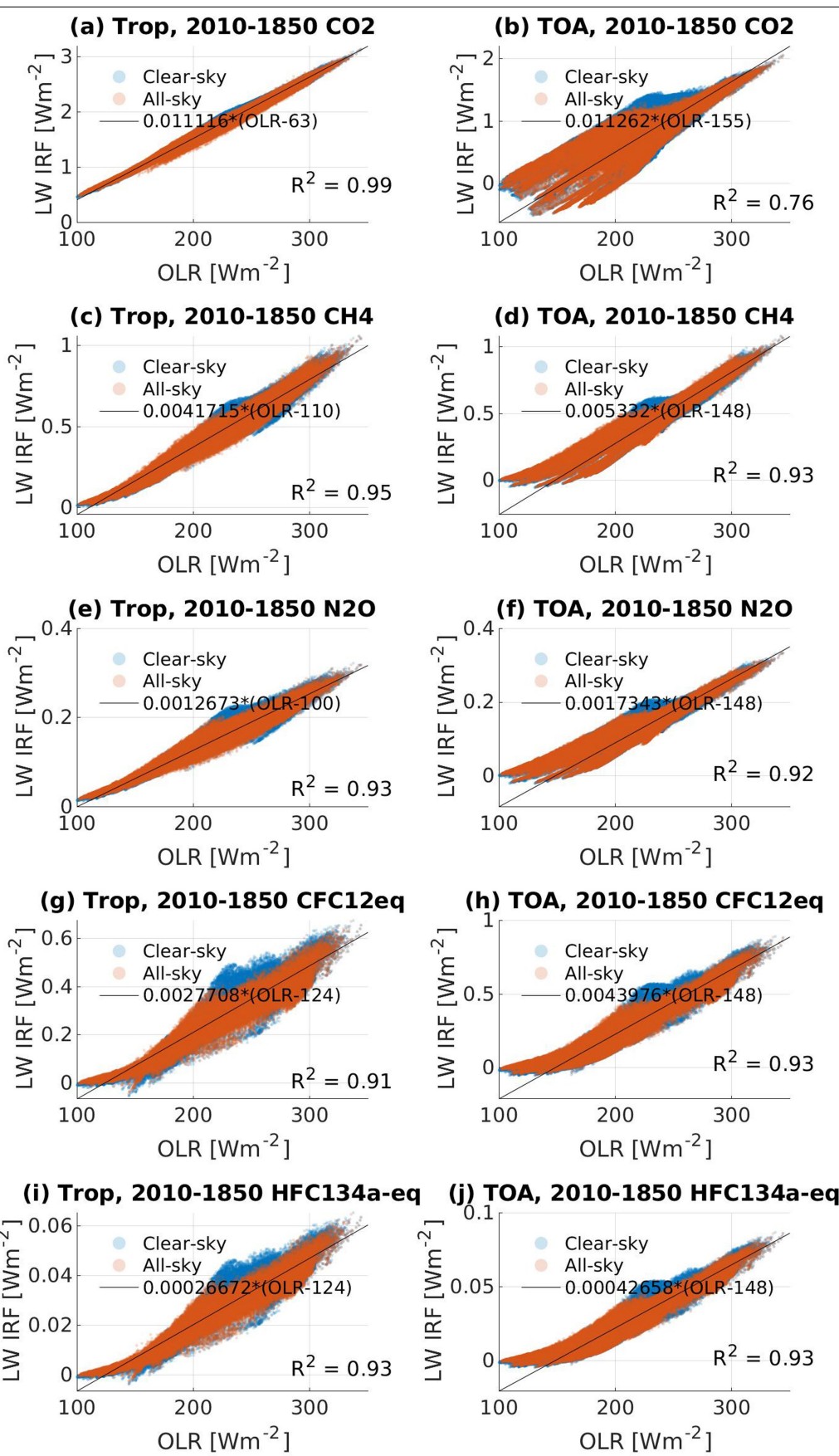

**Extended Data Fig. 3** | See next page for caption.

Extended Data Fig. 3 | Monthly, globally distributed LW IRF of the main WMGHGs exhibits a near-linear dependence on OLR. Monthly mean LW IRF as a function of OLR of the same location for $CO_2$ (**a**), $CH_4$ (**c**), $N_2O$ (**e**), CFC12-eq (**g**) and HFC134a-eq (**i**), each relative to pre-industrial (1850) concentrations. LW IRF$_{TOA}$ is shown in **b**, **d**, **f**, **h** and **j**, respectively. Linear regression coefficients (Table 1) are derived from clear-sky GRTcode–ERA5 (blue), whereas biases and $R^2$ values are evaluated using all-sky GRTcode–ERA5 (red). Each scatter point represents a monthly mean value for a grid cell from the GRTcode–ERA5 calculations at the year 2010.

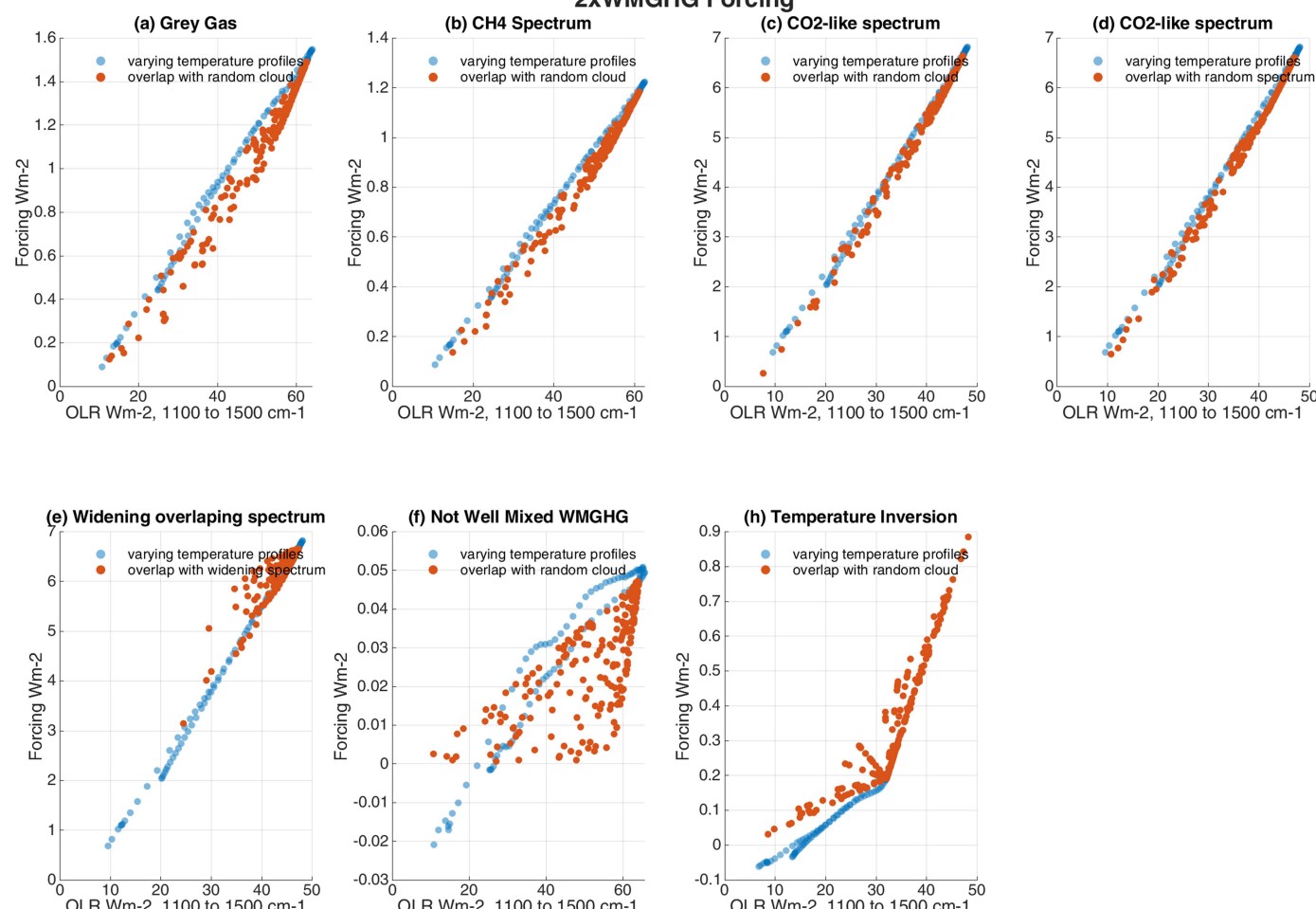

**Extended Data Fig. 4 | Idealized line-by-line experiments reveal the physical origin of the linear LW IRF–OLR relationship.** LW IRF is calculated for a doubling of a single WMGHG in the 1,100–1,500 cm$^{-1}$ spectral range, assuming no other absorbers. The radiative transfer equation is solved using Planck functions from zonal mean ERA5 temperature profiles, with an isothermal stratosphere. Blue markers show results using zonal mean annual mean temperatures (2.5° latitude spacing), representing diverse temperature conditions. Red markers show results with a fixed tropical profile and a randomly generated overlapping absorber (200 samples) in a random tropospheric layer to mimic the effects of overlapping gases and clouds. Panels **a** and **d**–**h** assume a spectrally uniform optical depth (greybody), panel **b** uses the CH$_4$ spectrum and panels **c**–**e** use a bell-shaped spectrum opaque near the tropopause, mimicking CO$_2$ band structure. The overlapping spectrum is uniform except in panels **c** and **d**, for which **c** uses a fixed random spectrum over the full range and **d** varies the spectral range. Panel **e** removes absorbers above 500 hPa and panel **f** introduces a tropospheric temperature inversion.

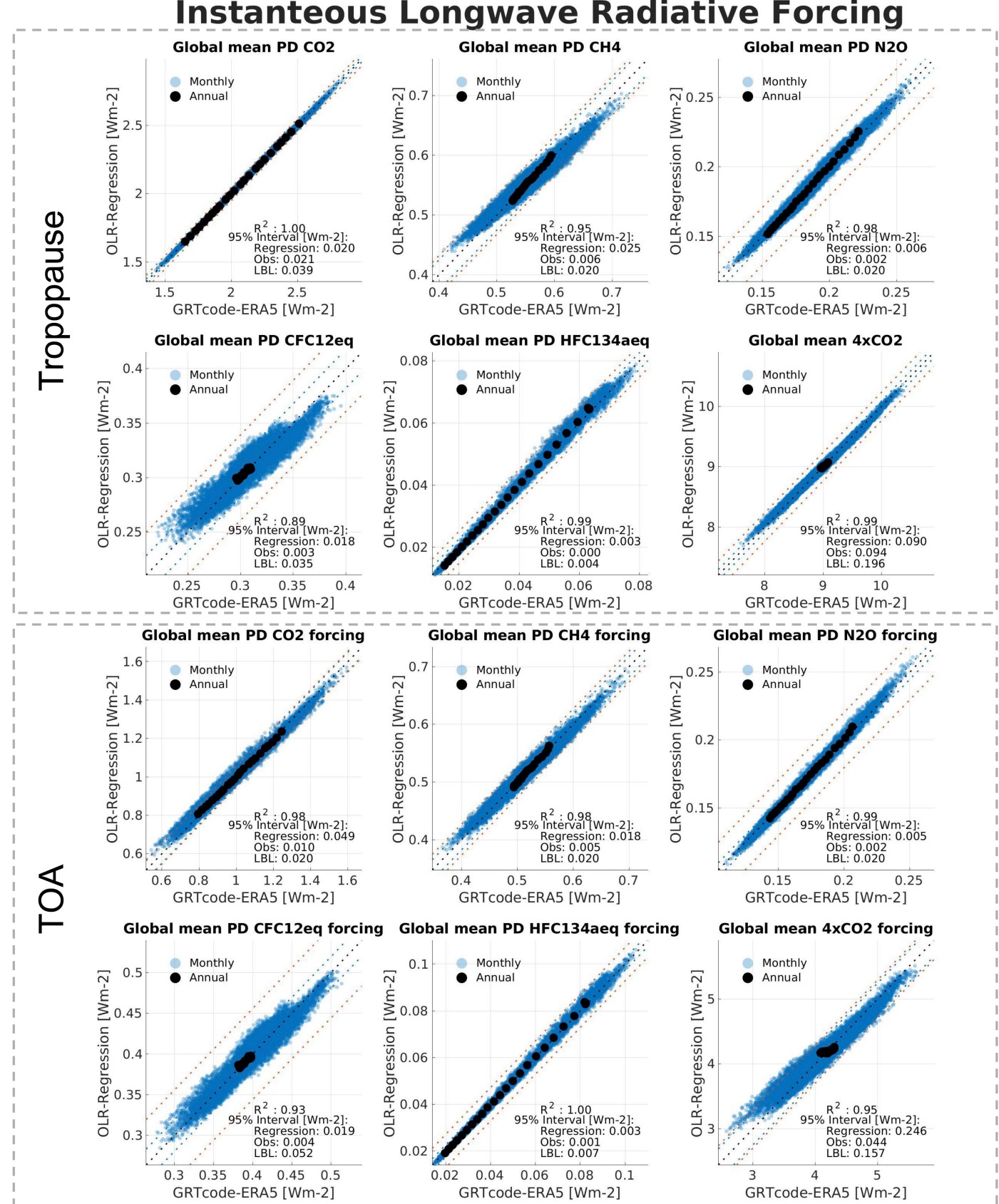

**Extended Data Fig. 5 | Evaluation of the OLR–regression method against line-by-line calculations.** Annual global mean (black) and monthly meridional mean (blue) LW IRF (upper panels; LW IRF$_{TOA}$ in the lower panels) from 2001–2024 are compared between GRTcode–ERA5 (x-axis) and the OLR–regression estimates (y-axis). Global mean biases, coefficients of determination ($R^2$) and the uncertainty ranges (95% confidence interval) associated with the regression method, clear-sky line-by-line calculations and OLR observations are indicated in each panel.

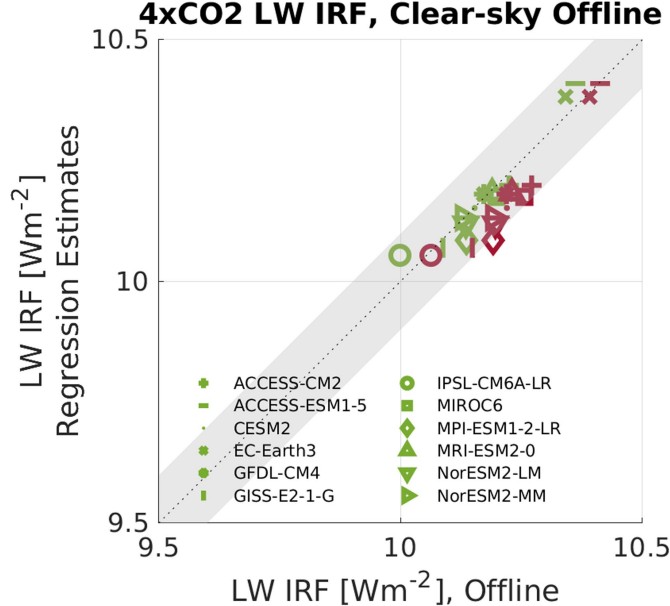

**Extended Data Fig. 6 | LW IRF estimated by the OLR–regression method compared with RTE-RRTMGP driven by background states of piClim-control (green) and piClim-4 × CO₂ (red) experiments.** The grey shading represents a 0.1 W m$^{-2}$ uncertainty range (at 95% confidence) of the OLR–regression method when applied to $4 \times CO_2$ LW IRF.

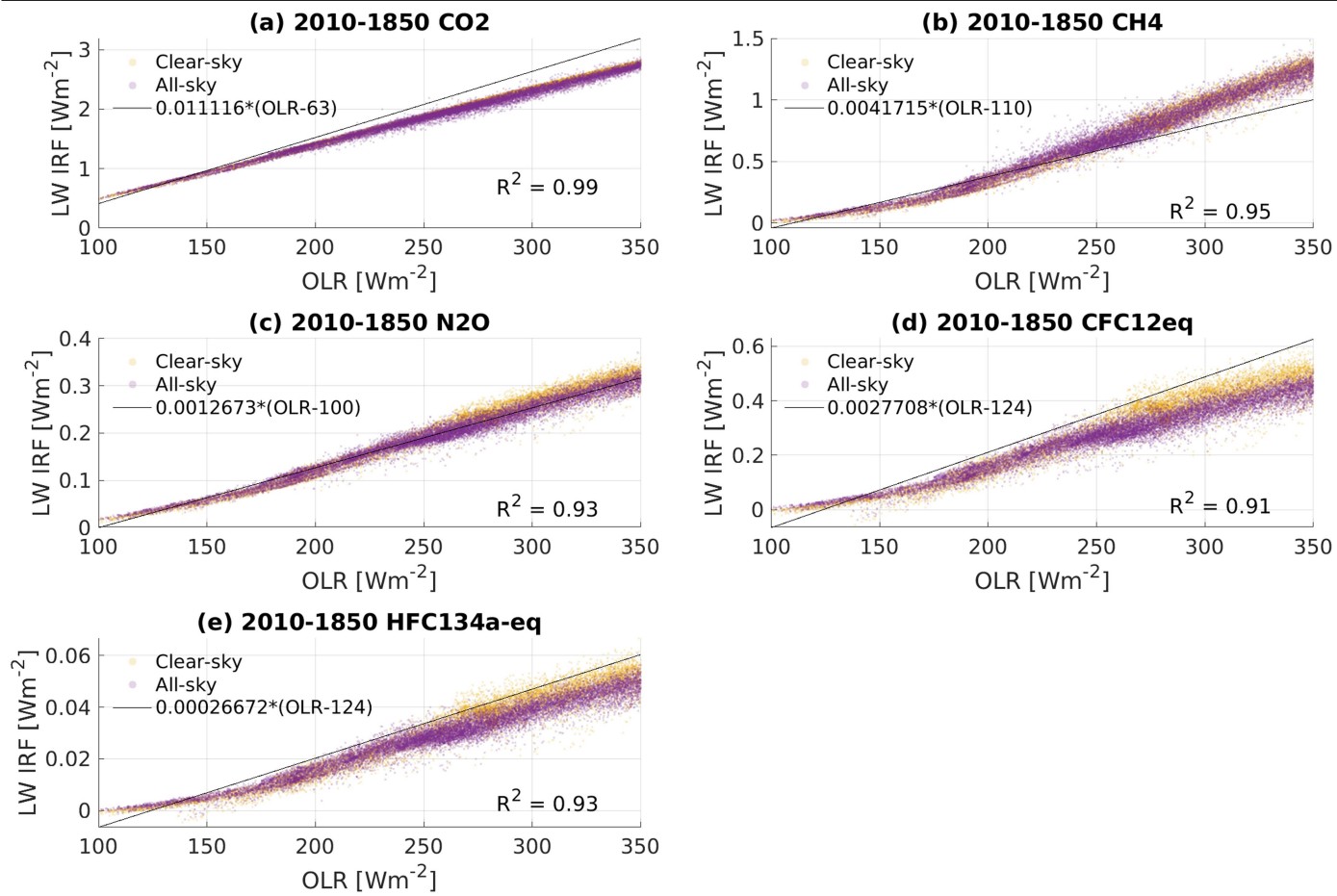

**Extended Data Fig. 7 | LW IRF of the main WMGHGs shows a near-linear dependence on OLR in the absence of water vapour.** Same as Extended Data Fig. 3 but removing water vapour from radiative transfer calculations.

**Extended Data Table 1 | Clear-sky LW IRF (W m$^{-2}$) from five line-by-line models[16,52-55] with respect to 100 profiles sampled from reanalysis products at year 2014**

| Scenario | 4AOP | ARTS | GRTcode | RFM | LBLRTM | Mean/STD | Regress | GRTcode-ERA5 |
|---|---|---|---|---|---|---|---|---|
| OLR baseline | 263.14 | 261.94 | 262.95 | 263.25 | 263.20 | **262.90/0.55** | - | 261.90 |
| LW IRF at 200 hPa tropopause | | | | | | | | |
| 2014-1850 $CO_2$ | 2.39 | 2.36 | 2.38 | 2.38 | 2.42 | **2.38/0.02** | **2.38** | 2.37 |
| 2014-1850 $CH_4$ | 0.64 | 0.64 | 0.64 | 0.64 | 0.64 | **0.64/0.01** | **0.65** | 0.65 |
| 2014-1850 $N_2O$ | 0.21 | 0.21 | 0.21 | 0.21 | 0.21 | **0.21/0.01** | **0.21** | 0.22 |
| 2014-1850 CFC+HFC | 0.42 | 0.42 | 0.42 | 0.41 | 0.38 | **0.41/0.02** | **0.43** | 0.43 |
| 4×$CO_2$ | 10.26 | 10.15 | 10.20 | 10.23 | 10.40 | **10.25/0.10** | **10.22** | 10.17 |
| LW IRF$_{TOA}$ | | | | | | | | |
| 2014-1850 $CO_2$ | 1.30 | 1.31 | 1.29 | 1.29 | 1.32 | **1.30/0.01** | **1.31** | 1.29 |
| 2014-1850 $CH_4$ | 0.62 | 0.61 | 0.62 | 0.62 | 0.61 | **0.61/0.01** | **0.62** | 0.62 |
| 2014-1850 $N_2O$ | 0.21 | 0.21 | 0.20 | 0.21 | 0.21 | **0.21/0.01** | **0.21** | 0.21 |
| 2014-1850 CFC+HFC | 0.57 | 0.56 | 0.56 | 0.55 | 0.50 | **0.55/0.03** | **0.56** | 0.56 |
| 4×$CO_2$ | 5.40 | 5.45 | 5.32 | 5.37 | 5.51 | **5.41/0.08** | **5.38** | 5.33 |

Two times the standard deviations across these models are used to infer 95% confidence interval in the line-by-line calculation of IRF. The OLR–regression method ('Regress') is applied using multi-model mean clear-sky OLR and gas concentrations from ref. 16. The GRTcode–ERA5 results for the global mean of 2014 are listed on the basis of an updated gas concentration dataset following CMIP7 (ref. 13).

**Extended Data Table 2 | All-sky LW IRF (Wm$^{-2}$) caused by increases in the main WMGHGs from 1850 to 2014**

| Source | Gas | Net SARF [Wm$^{-2}$] | LW IRF [Wm$^{-2}$] | Constrained LW IRF [Wm$^{-2}$] |
|---|---|---|---|---|
| Etminan et al. (2016)[15] | CO$_2$ | 1.806 | 2.059 | $2.096 \pm 0.05$ |
| | CH$_4$ | 0.566 | 0.521 | $0.548 \pm 0.02$ |
| | N$_2$O | 0.175 | 0.173 | $0.188 \pm 0.006$ |
| IPCC AR6 Ch.7 Supp.[1] | CFC12-eq | 0.358 | 0.322 | $0.304 \pm 0.039$ |
| | HFC134a-3q | 0.040 | 0.036 | $0.038 \pm 0.005$ |
| | | **Net IRF [Wm$^{-2}$]** | | |
| Byrne & Goldblatt (2014)[73] | CO$_2$ | 1.830 | 1.922 | $2.096 \pm 0.05$ |
| | CH$_4$ | 0.525 | 0.515 | $0.548 \pm 0.02$ |
| | N$_2$O | 0.198 | 0.198 | $0.188 \pm 0.006$ |

Estimates from ref. 15 and IPCC AR6 (ref. 1) are converted from net stratospherically adjusted radiative forcing (SARF) using conversion ratios in Extended Data Table 3 (second column). Estimates from ref. 73 are converted from net IRF using Extended Data Table 3 (third column). CERES-constrained LW IRF based on the OLR–regression methods are also shown for comparison.

**Extended Data Table 3 | Conversion ratios from SARF or net IRF to LW IRF for the main WMGHGs**

| Gas | LW IRF / Net SARF | LW IRF / Net IRF |
|---|---|---|
| $CO_2$[15,45] | 1.14 | 1.05 |
| $CH_4$[15,45] | 0.92 | 0.98 |
| $N_2O$[15,16,45] | 0.99 | 1.00 |
| CFC[3,74] | 0.90 | – |
| HFC[3,74] | 0.90 | – |

References are listed in the left column for each gas[3,15,16,45,74].

**Extended Data Table 4 | Radiation flux diagnostics (W m⁻²) based on online double-call radiation of AMIP of six models, including BCC-CSM2-MR[77], CNRM-CM6-1 (ref. 78), IPSL-CM6A-LR[79], MIROC5 (ref. 80), GFDL-CM4 (ref. 81) and HadGEM3-GC31-LL[82]**

| Variable | BCC | CNRM | IPSL | MIROC5 | GFDL | HadGEM | Mean / Std |
|---|---|---|---|---|---|---|---|
| $F_{\mathrm{fsst}}$ | 7.18 | 8.44 | 8.18 | 8.81 | 8.23 | 7.73 | 8.10 / 0.57 |
| OLR | 238.65 | 238.20 | 237.94 | 235.89 | 238.54 | 241.04 | 238.38 / 1.65 |
| LW IRF$_{\mathrm{TOA}}$ | 6.72 | 4.23 | 4.09 | 4.39 | 3.50 | 4.08 | 4.50 / 1.13 |
| $F_{\mathrm{fsst}} = $ LW IRF + SW IRF + Adj | | | | | | | |
| LW IRF | 7.62 | 9.19 | 9.13 | 9.55 | 8.75 | 8.67 | 8.82 / 0.67 |
| SW IRF | -0.14 | -0.69 | -0.84 | -0.56 | -0.48 | -0.39 | -0.52 / 0.24 |
| Adj | -0.30 | -0.06 | -0.11 | -0.17 | -0.04 | -0.55 | -0.20 / 0.19 |
| LW IRF (Regression) | 8.90 | 8.88 | 8.86 | 8.76 | 8.90 | 9.02 | 8.89 / 0.08 |
| Radiation bias | 1.28 | -0.31 | -0.27 | -0.79 | 0.15 | 0.35 | 0.07 / 0.71 |
| $F_{\mathrm{fsst}}$ (bias corrected) | 8.46 | 8.13 | 7.91 | 8.03 | 8.38 | 8.08 | 8.16 / 0.21 |

$F_{\mathrm{fsst}}$ is calculated as the net TOA radiation difference between AMIP and AMIP-4×CO$_2$ of the same model.