## [Peer Review file · Nature]

A Strong Constraint on Radiative Forcing of Well-mixed Greenhouse Gases

Corresponding Author: Dr Jing Feng

Version 0:

Reviewer comments:

Referee #1

(Remarks to the Author)

This manuscript seeks to establish stronger constraints on all-sky longwave instantaneous radiative forcing (the change in outgoing longwave radiation at the top of the atmosphere induced by changes in atmospheric composition). The forcing is known to depend on the state of the atmosphere including the distribution of clouds. The authors perform high-accuracy, high-cost line-by-line calculations on 24 years worth of global atmospheric states as represented by monthly-mean reanalysis fields, including calculations with a set of five greenhouse gas concentrations set to pre-industrial values. Forcing is shown to depend linearly on outgoing longwave radiation (OLR) across a wide range of OLR values in both the presence and absence of clouds. Linear regression models are constructed that allow forcing due to each of the five gases to be reconstructed from observations of state (as represented by a reanalysis) or from observations of OLR, extending the time series by a decade relative to the last available benchmarks. The regression is also used to isolate errors in climate model representations of radiative transfer and to estimate the impact of the evolving (2000-2024) atmospheric state on radiative forcing by quadrupled concentrations of CO₂ relative to pre-industrial values.

The central insight - that radiative forcing by well-mixed greenhouse gases is linearly related to state-dependent OLR - has never been expressed so clearly to my knowledge. This is a neat idea. It also makes good sense, since forcing is a small perturbation on OLR. Exploiting the small-perturbation nature of forcing to estimate all-sky radiative forcing is a clever way around the otherwise high-dimensional problem of including clouds.

But such a simple idea would benefit from a more principled or theoretical explanation rather than emerging from staggering amounts of computation, with regression coefficients of unknown sensitivity that can't be traced to first principles. The practical implications of the regression offer modest real-world benefit: the estimates of forcing from reanalysis amount to using the clear-sky/all-sky ratio of forcing from ERA to estimate all-sky forcing following [14], the uncertainty of such estimates (as distinct from variations across climate models) is already quite low, and a more careful treatment of conditional uncertainty would be required to fully understand the degree to which estimates are actually refined. The manuscript would benefit too from substantial improvements in clarity, precision, and narrative flow.

The linear relationships between OLR and radiative forcing, which holds better at the tropopause than at the TOA (Figs S3, S4), could be interpreted in two contexts. One could view this as an "emergent constraint" (e.g. doi:10.1038/s41558-019-0436-6) - but such constraints have been useful when the variability across models is much larger than is the case for real-world IRF, and have rarely turned out to be robust or strong. The other approach would be identify some fundamental reason why OLR and forcing should be linear. That forcing is a small perturbation is a start but it seems likely that some consideration of spectral variation will be needed. (It seems possible, for example, that masking by stratospheric CO₂ as described in 10.1029/2023MS003729 would explain why linearity holds better at the tropopause than the TOA). In the absence of deeper understanding claims that the linear relationships provide "strong constraints" are unfounded.

The manuscript covers two somewhat disparate topics: a) the temporal evolution from 2020-2025 of instantaneous radiative

forcing (IRF) by CO₂, CH₄, N₂O, and two sets of halocarbons and b) the role of parameterization error in explaining the range of IRF across a subset of models contributing to CMIP. These are somewhat interwoven in that Figure 3 and the relevant text, on the latter subject, comes between figure 1-2 and figure 4, which address the former. The confusion also arises in the introduction, where the material describing the precision with which real-world IRF is known veers into a discussion of variability across earth system models at line 61. These are distinct subject, linked here only by the use of the same regression technique to explore them. The manuscript would benefit from making the conceptual distinction between these questions more clear and not, for example, equating diversity across ESMs with uncertainty (line 62). It is not clear that the two subjects belong in the same manuscript.

More detailed comments:

Errors in “humidity and cloud fields in reanalyses and ESMs” (line 60) have both very different causes and different implications for computing IRF. On the one hand the IRF to which a model is subject depends only on the model state - the uncertainty (line 62) arises only from errors in approximation. Reanalyses might well be biased in ways that affect OLR and forcing calculations but the cited literature does not support this claim - and indeed the authors are happy to make calculations based on ERA5. Should these not be trusted after all?

The paragraph stating on line 68 does set the stage for the work that follows. It seems possible that the authors want to anticipate applying their regression results to CERES observations of broadband irradiance; this might be made more clear by omitting some of the more general description.

The assertion on lines 77-79 is more dramatic than warranted. The uncertainty in estimates of clear-sky radiation is of order 1%, as [14] notes; applying the GRT code or any line-by-line model to greenhouse gas concentrations from 2000-2024 using the small number of samples from that paper would be a much simpler exercise than the present manuscript undertakes. If the statement is meant to apply to clouds the authors might be more precise.

How are clouds treated in the ERALBL computations?

The value of IRF as measured at the tropopause results in part from the strong adjustment of stratospheric temperature to the increases in CO₂ that constitute the majority of 20th and 31st—century forcing. Are the authors aware of evidence that this also holds for other greenhouse gases for which the stratospheric adjustment is small?

Lines 114-121 are not consistent with my reading of [14]. Their Table 3 shows a range of all-sky IRF/clear-sky IRF at the top of atmosphere across models as being something like 0.735-0.767. The value reported for LBLERA is 0.734 - which seems consistent, not inconsistent, with table 3.

Lines 122-128 seem irrelevant. The simplified expressions for radiative forcing described here are not intended as benchmarks in the same sense as [14] or the present work.

Line 129-130: Clouds are not constrained by any observations in ERA5.

The meaning of lines 151-152 is not clear.

Line 158 reports that regression coefficients between OLR and forcing derived for clear skies hold equally well for profiles including clouds. Is this true for some fundamental reason, or are the errors incurred in the translation just small because forcing is a small fraction of OLR?

That the regression methods can reconstruct forcing from observations of OLR is a neat trick. It is less clear what differences between LBLERA, the regression model applied to LBLERA (which agrees by construction), and regression applied to observations tells us about the IRF to which the world is subject, especially given that differences are of order 0.025 W/m² out of 9 W/m².

(Remarks on code availability)

Referee #2

(Remarks to the Author)

Summary:

The manuscript “A Strong Constraint on Radiative Forcing of Well-mixed Greenhouse Gases” by Feng et al. provides the first benchmark calculation of the longwave instantaneous all-sky forcing of well-mixed greenhouse gases (WMGHG), both at the top of the atmosphere and the tropopause. To achieve this, they use a computationally optimized but very accurate line-by-line radiative transfer model based on atmospheric input from reanalysis data over a period of more than 20 years. The authors use this benchmark to derive an empirical regression model utilizing the strong relationship between outgoing longwave radiation (OLR) and radiative forcing that is remarkably similar between clear-sky and all-sky conditions. This regression model in turn is applied to observed OLR for both clear-sky and all-sky conditions, yielding very good agreement with the line-by-line simulations. This way, they incorporate the effects of clouds and changing temperatures on the forcing and provide a useful tool to diagnose instantaneous radiative forcing from coupled climate models.

The study addresses a central and relevant question in climate science: Despite a generally very good understanding of atmospheric radiative transfer, uncertainties in WMGHG forcing have been persistent due to limitations of radiation parameterizations, biases in the atmospheric states of climate models, as well as the effects of clouds and of changing global temperatures. Whereas previous studies on this topic have addressed some of these challenges, a major advance in this work is that they simultaneously address all these challenges. By combining the advantages of line-by-line radiative transfer simulations, reanalysis data, and satellite observations, this work better constrains the magnitude of the longwave instantaneous all-sky forcing of the most important greenhouse gases.

The manuscript is clear and well-written. The authors use a highly accurate radiative transfer code, combined with a state-of-the-art reanalysis dataset. They derive a regression model that is then applied to a well-established observational satellite product, which is used to demonstrate the robustness of their approach. They advance the field by providing a better constraint on the longwave instantaneous all-sky WMGHG forcing and a tool to diagnose it from Earth system models. Their findings are highly relevant for the atmospheric radiative transfer community but a better constraint on the radiative forcing of WMGHG also has broader implications for climate science more generally. In my opinion, the manuscript is therefore suited for publication in Nature. I enjoyed reading this manuscript and only have some minor comments.

Florian Roemer
ETH Zurich

General comments:

1. While the authors provide good justifications to focus on the longwave forcing component (much larger magnitude and higher uncertainty), the manuscript would be even stronger if it also discussed implications (or possible extensions) of this work for the shortwave component. Building on the strong relationship between outgoing longwave radiation and longwave forcing, one could, for example, imagine a similar relationship between reflected shortwave radiation and the shortwave forcing. Have the authors tested this? Or is there a reason why it was not feasible in this setup? If so, what would be necessary to enable such an extension to the shortwave domain? A discussion of this would make the manuscript even more useful to the community.
2. The authors mainly analyze the forcing at the tropopause. While this is made clear throughout the manuscript, it is not clear from the abstract and conclusion, where it is simply referred to as "instantaneous forcing". In my opinion, this undersells the achievements of this study, because the tropopause forcing is more closely linked to the effective forcing after stratospheric adjustments and thus the more relevant quantity for surface warming compared with the instantaneous forcing at the top of the atmosphere. I suggest also mentioning this point in the conclusions.
3. The regridding of the ERA5 data to a much coarser resolution (around 250km rather than the native 31km) is understandable given the expensive nature of line-by-line radiative transfer. However, it should not just be mentioned in passing in the methods but rather be made more transparent by mentioning it in the main text and briefly discussing the limitations of this approximation, particularly on cloud effects (the latter can be done in the methods).

Specific comments:

- Line 55-56: The sentence should start with an upper-case "L".
- Line 72: It should either say "often exceeding" or "that often exceed".
- Lines 167-169: Please provide more details on how exactly the confidence range is estimated from the meridional means (either here or in the methods).
- Lines 179-180: The part of the sentence after the semicolon is presumably missing a verb.
- Line 246: Does this refer to the method described in lines 167-169? Please clarify.
- Lines 250-252: This sentence is confusing to me, as it appears to imply that the forcing increase is caused by the surface warming (i.e., the response). Please rephrase.
- Line 269: The reference should presumably be to Fig. 3c?
- Line 306: An "and" is missing after the last comma.
- Line 325: "annual-mean, monthly-mean": does this mean that both are used? Please rephrase.
- Fig. 2: Please explain the error bar in the legend and caption.
- Fig. 3: Please explain the meaning of the colored shadings in the legend and caption.

(Remarks on code availability)

The provided code is generally well-documented, including a README file describing how to build GRTcode and a Guide file outlining the steps to run GRTcode for the reanalysis input. As far as I can tell, there does not seem to be any documentation how the data is post-processed to produce the figures in the manuscript.

Referee #3

(Remarks to the Author)

This paper seeks to definitively establish benchmark calculations of the radiative forcing of greenhouse gases over the last several decades. It does so by performing a massive number of radiative transfer calculations with realistic atmospheric

states as they occurred in over the last several decades, so that the calculations are fully aware of the atmospheric state and how that impacts these calculations.

The paper represents an impressive result technically because of the sheer volume of radiative transfer calculations necessary to achieve this paper's results. A vast amount of computational resources were expended for this paper in order to create the definitive benchmark of greenhouse gas radiative forcing calculations. It may be acceptable for publication with major revisions.

There are several revision needs. First, it is that it is unclear if the impressive technical results fundamentally alter the scientific community's understanding of the radiative forcing of greenhouse gases, or if the calculations are just about reducing uncertainty in the quantity. Specifically, do they show that the community needs to keep doing this this level of calculation moving forward?

In order for the manuscript to rise to be of broad interest, the authors need to discuss in detail if this comprehensive result alters our understanding of radiative forcing of these gases in general, and if it alters how the community should consider what is sufficient for estimating greenhouse gas radiative forcing. The authors do compare their epic calculations to established GHG RF formulae and it appears that their result does alter and upward-revise the understanding of greenhouse gas RF based on a small number of calculations, e.g., in Etmnan et al, 2016 and Pincus et al, 2020? Why is this the case? Does a more complete sampling of the atmosphere need be included in the formulae? Were the impacts of clouds underestimated in previous simple formulae and can they be represented in formulae moving forward? Is an underestimation guaranteed unless there are large-scale calculations such as those presented here? Or can better sampling of atmospheric states for formulae generation be done? Given the calculations that the authors performed, a quick analysis of the data already in hand would answer this question and I recommend the authors do that analysis.

The reason it is important to connect these results back to simple formulae for each gas species is that there are large communities that use such formulae for, to name just a few examples, adaptation, mitigation, and resilience planning. If a simple formula is insufficient because it is low-biased currently, or because it will always be low-biased due to its simplicity, then many communities need to know about it and know that achieving accurate GHG RF estimates for applications purposes is not straightforward.

Alternatively, can the community have confidence that a simple formula or formulae is sufficient, perhaps with tweaking from calculations like these. Or can the authors say that the level of calculation they performed needs to be done once but not again? Or is that problematic because the atmospheric state, including thermodynamics and clouds (and its response to forcing), will continue to dominate the RF error budget?

Or is spectroscopy now and into the future the driver of uncertainty?

There is much that can be gained in understanding from such an epic set of calculations, and the paper needs revisions, including an assessment of whether tweaks to simple formulae will be sufficient for assessing greenhouse gas RF moving forward, especially for individual gases.

Finally, I'll comment on the point the make authors showing that LW IRF is related to clear-sky OLR. This is an intriguing result. From this, the take-away message is that as long as the remote sensing community continues to accurately measure this quantity with satellite instrumentation, then the larger scientific community can use their linear regression to derive a metric of LW IRF for all greenhouse gases. Are the reported accuracies and precision values of satellite-derived OLR products sufficient to do this and are their implications for the long-term on-orbit stability that instruments and associated OLR products need to achieve?

(Remarks on code availability)

A brief review of the code indicates that it is thorough and usable for the community. I did not install or run the code, but it does have a README file.

Version 1:

Reviewer comments:

Referee #1

(Remarks to the Author)

The revision has clarified some points but the manuscript structure remains essentially unchanged, so that the work represents (as noted by reviewer 3) a new approach for computing radiative forcing based on outgoing longwave radiation without any change in understanding or conception. The approach might provide a higher-accuracy estimate of radiative forcing than does computation with a finite number of profiles [13, 14, 23], especially with respect to cloudy atmospheres, and so might have value in the specialist literature. It's hard to see how modest improvements in the accuracy of an already well-constrained problem, based on massive computation but absent physical arguments, warrant publication in a high-impact journal even in the absence of confusion and conflicts with previous work as outlined below.

The manuscript introduces unnecessary confusion on a number of points.

- The value of determining climate model-specific radiative forcing is that it helps disentangle variations in forcing from variations in response (feedbacks). The forcing to which each model is subject includes any errors in model-specific treatments of radiative transfer, so it's unclear what is gained by "constraining" or "benchmarking" radiative forcing estimates from models contributing to CMIP6. (The models are in no sense an estimate of the uncertainty in forcing to which the earth is subject.) The roles of state dependence and model error in determining the diversity of model forcing are transparently available by applying off-line calculations (as is done here and in [20]); it's not clear what value the use of a regression model adds.

- The focus on instantaneous radiative forcing at the tropopause is not consistent with the last several decades of thinking. Interest in radiative forcing at the troposphere (e.g. Hansen et al. 1999, doi:10.1029/96JD03436) arose before stratospheric cooling (which really affects only CO₂, as [34] shows) was understood as one of many possible radiative adjustments. It's interesting that radiative forcing is more linear in OLR at the tropopause than at the TOA but this does not make the former more relevant than the latter.

- The regression models used for predicting forcing from OLR depend on concentration in non-linear ways (Tables 1 and 2). Why should a linear relationship hold across more than W/m² in OLR but not across modest changes in the concentration of greenhouse gases?

The manuscript conflicts with several existing results but does not mention or attempt to reconcile these conflicts. In particular:

- Reference 20 reports that much of the variability in radiative forcing due to CO₂ arises from differences in stratospheric temperature; here the authors claim (lines 236-237) that the calculation of LW IRF is the dominant cause of spread.

- Reference 27 reports that radiative forcing by halocarbons is strictly linear in concentration across a much wider range of concentrations than is explored here, but Tables and 1 use quadratic forms.

(Remarks on code availability)

Referee #2

(Remarks to the Author)

The authors have satisfactorily addressed all the points raised in my first review. I have no further comments.

Florian Roemer
ETH Zurich

Addendum:

I think that some of referee 1's points are valid, but there are other points I disagree with.

I do agree that the major contribution of this work is methodological with smaller contributions to physical understanding. However, due to their innovative methodology they better constrain the all-sky long wave radiative forcing of greenhouse gases, simultaneously addressing most of the existing uncertainties caused by parameterization errors, climate model biases, and cloud effects. Thus, I would still argue that the resulting constraint on this forcing is relevant beyond the specialist literature, whether that is in Nature or another journal is hard for me to judge.

To his specific points on confusion:

- I agree that the spread in radiative forcing estimates from climate models is different from the uncertainty in the real-world radiative forcing, and this should be mentioned in the manuscript. Nevertheless, climate models play an important role in assessing the value of real-world radiative forcing, so narrowing the spread is certainly valuable.
- I disagree with his criticism of using the instantaneous radiative forcing (IRF) at the tropopause. There is good reason to look at this quantity rather than IRF at the top of the atmosphere (TOA). Because of stratospheric adjustments, there are substantial differences between instantaneous and effective forcing at TOA, the latter being the relevant quantity for the climate response. The use of the IRF at the tropopause partially circumvents this problem, as its value is closer to the effective forcing at TOA than is the instantaneous forcing at TOA (see for example <https://doi.org/10.1175/JCLI-D-19-0756.1>).
- I am not entirely sure what referee 1 means by this point. The logarithmic dependency of forcing on greenhouse gas concentration is well-established for optically thick gases such as CO₂ in Earth's atmosphere. For the non-CO₂ gases, I do think a more detailed explanation for the choice of model is warranted (see comment below).

To his specific points on conflicts with existing literature:

- I don't see the conflict with reference 20, as they only discuss IRF at TOA whose uncertainty is dominated by biases in stratospheric temperatures. In contrast, this manuscript claims that most of the uncertainty in effective forcing at TOA comes from the IRF at the tropopause. I do think that it would strengthen the manuscript to contrast the different sources of uncertainty in ERF and IRF at TOA, but I don't see how this result conflicts with existing literature.
- I do see the need for the authors to explain their use of a quadratic model for halocarbons in contrast to the linear relationship found by reference 27.

(Remarks on code availability)

I did not run the provided code, but it now also includes scripts to postprocess and plot the data which should make it easier to reproduce the results.

Referee #3

(Remarks to the Author)

I thank the authors for their extensive revisions.

The revised manuscript is much improved from the original submission and satisfactorily addresses the points I raised and appears to address the issues raised by the other reviewers too. Furthermore, the importance and impact of the manuscript is now clear. It also provides a strong motivation for the scientific community to (continue to) measure OLR accurately to track GHG RF.

My one minor remaining comment is that given the findings of this manuscript, I would recommend that the authors reference the issues/concerns raised in Soden et al, 2018, (<https://www.science.org/doi/10.1126/science.aau1864>) and then comment in the manuscript's conclusion about how/whether their work addresses those issues or if there is still more work to be done.

(Remarks on code availability)

While I have not run their code, I have previewed it. The code is straightforward to run and interpret and appears to be consistent with and supportive of the findings in their manuscript.

Response to Reviewer Comments

We would like to thank the editors and reviewers for their constructive comments, which have greatly helped improve the clarity and overall quality of this manuscript. We recognized that discussions of the broader applicability of our approach and the supporting evidence were not made sufficiently explicit or clearly linked to the main text in the initial submission. While the main results and conclusions remain consistent with those of the original version, we have substantially strengthened the manuscript in the following aspects:

- We have expanded the explanation of the linear relationship between IRF and OLR in the *Methods* and added idealized experiments (L418-465, Extended Data Figure 5). Together with ERA5LBL calculations (clear-sky no water vapor, all-sky no water vapor, clear-sky, and all-sky cases), these experiments support our key claim that as long as the global-mean climate state lies within the broad range of seasonal and spatial variability of present-day Earth, the benchmark method proposed here should generally hold.
- To facilitate broader community use, Tables 1–2 and the formulation of the regression slope have been refined to ensure a consistent representation of slope dependence on WMGHG concentration (from preindustrial to four times the present-day level). For concentrations beyond this range, we have added guidance on how to apply the regression method to rescale results from existing approaches, improving their suitability for climate states outside their original sampling conditions. This revision slightly modifies the expression of Eq. (3) but does not affect the estimated IRF values.
- We have added a short discussion of the broader applications of the method in the *Methods* (L500-520). Scripts for direct implementation of the regression approach have been uploaded to Code Ocean and Zenodo. Discussions of uncertainties have also been made clearer and more explicit (L466-498).

We have also carefully revised the manuscript for clarity and consistency throughout. Detailed responses to individual comments are provided below, with line numbers referring to the revised version.

Responses to Reviewer 1

1. *This manuscript seeks to establish stronger constraints on all-sky longwave instantaneous radiative forcing (the change in outgoing longwave radiation at the top of the atmosphere induced by changes in atmospheric composition). The forcing is known to depend on the state of the atmosphere including the distribution of clouds. The authors perform high-accuracy, high-cost line-by-line calculations on 24 years worth of global atmospheric states as represented by monthly-mean reanalysis fields, including calculations with a set of five greenhouse gas concentrations set to pre-industrial values. Forcing is shown to depend linearly on outgoing longwave radiation (OLR) across a wide range of OLR values in both the presence and absence of clouds. Linear regression models are constructed that allow forcing due to each of the five gases to be*

reconstructed from observations of state (as represented by a reanalysis) or from observations of OLR, extending the time series by a decade relative to the last available benchmarks. The regression is also used to isolate errors in climate model representations of radiative transfer and to estimate the impact of the evolving (2000-2024) atmospheric state on radiative forcing by quadrupled concentrations of CO₂ relative to pre-industrial values.

The central insight - that radiative forcing by well-mixed greenhouse gases is linearly related to state-dependent OLR - has never been expressed so clearly to my knowledge. This is a neat idea. It also makes good sense, since forcing is a small perturbation on OLR. Exploiting the small-perturbation nature of forcing to estimate all-sky radiative forcing is a clever way around the otherwise high-dimensional problem of including clouds.

But such a simple idea would benefit from a more principled or theoretical explanation rather than emerging from staggering amounts of computation, with regression coefficients of unknown sensitivity that can't be traced to first principles. The practical implications of the regression offer modest real-world benefit: the estimates of forcing from reanalysis amount to using the clear-sky/all-sky ratio of forcing from ERA to estimate all-sky forcing following [14], the uncertainty of such estimates (as distinct from variations across climate models) is already quite low, and a more careful treatment of conditional uncertainty would be required to fully understand the degree to which estimates are actually refined. The manuscript would benefit too from substantial improvements in clarity, precision, and narrative flow.

We thank the reviewer for an insightful summary and constructive comments. In response to the general comments, we have 1) added more explanation of the linearity using an idealized experiment setup in the Method Section to demonstrate essential conditions for the proposed linear regression method to hold, 2) substantially improve the introduction and the logic flow throughout, 3) improve clarity in how the regression method should be implemented by a broader community.

2. *The linear relationships between OLR and radiative forcing, which holds better at the tropopause than at the TOA (Figs S3, S4), could be interpreted in two contexts. One could view this as an “emergent constraint” (e.g. doi:10.1038/s41558-019-0436-6) - but such constraints have been useful when the variability across models is much larger than is the case for real-world IRF, and have rarely turned out to be robust or strong. The other approach would be identify some fundamental reason why OLR and forcing should be linear. That forcing is a small perturbation is a start but it seems likely that some consideration of spectral variation will be needed. (It seems possible, for example, that masking by stratospheric CO₂ as described in 10.1029/2023MS003729 would explain why linearity holds better at the tropopause than the TOA). In the absence of deeper understanding claims that the linear relationships provide “strong constraints” are unfounded.*

We agree that a spectral explanation is essential to understand why IRF scales linearly with OLR. A discussion has been added to the Method section as:

Idealized line-by-line experiments indicate that the linear OLR–IRF relationship arises from basic radiative-transfer physics. Using the ERA5 zonal-mean temperature profile with an isothermal stratosphere and a vertically uniform **graybody absorber**, IRF_{TOA} is nearly linear with OLR. This linearity largely persists when **random** overlapping absorbers of varying optical depths and altitudes are added (Extended Data Figure 5a–d) and breaks down only when the spectral width of the overlapping absorber varies with height (Extended Data Fig. 5e), the gas is not well mixed (Extended Data Fig. 5f), or strong temperature inversions occur (Extended Data Fig. 5g). These experiments demonstrate that a near-linear LW IRF–OLR relationship is **a generic consequence of additive monochromatic radiative fluxes**, and thus expected for all well-mixed greenhouse gases under Earth-like conditions.

When overlapping with clouds, **realistic absorption spectra (e.g., CO_2 , CH_4) preserve the OLR–IRF relationship even more effectively than a gray absorber** (Extended Data Fig. 5b–c). ERA5LBL simulations further show that **clouds have negligible influence on the relationship for every major WMGHG (Extended Data Figs. 3 and 9), with or without water vapor**. The **small deviations observed in realistic atmospheres primarily arise from water vapor spectral overlap**, which slightly alters the regression slope (Extended Data Fig. 9).

In the ERA5LBL results, the only exception to this linear relationship is the CO_2 IRF_{TOA} . This contrasts with the enhanced linearity found in the idealized experiment (Extended Data Figure 5c) that assumed an isothermal stratosphere. The deviation arises because LW IRF_{TOA} is sensitive to stratospheric temperature and lapse rate [Jeevanjee et al., 2021, He et al., 2023, Chen et al., 2024], both of which exhibit strong seasonal variability driven by the Brewer–Dobson circulation [Young et al., 2011]. In contrast, **OLR is insensitive to temperatures above the tropopause, as these layers only contribute to TOA in the strong absorption bands of CO_2 and O_3 and are partly masked by stratospheric CO_2 [Cronin and Dutta, 2023].**

...

The LW IRF used in this study, defined at the tropopause, is largely **insensitive to tropopause or stratospheric temperatures, similar to OLR**. For CO_2 , **the LW IRF from upwelling fluxes at the tropopause shares the same sensitivity to tropopause temperature as the downwelling component [Jeevanjee et al., 2021], effectively canceling the dependence on tropopause temperature**. For other WMGHGs, whose stratospheric absorption is weak, defining IRF at the TOA or tropopause produces small differences. Because the downwelling fluxes at the tropopause are mainly controlled by the local temperature, where WMGHGs are most abundant, the LW IRF is largely unaffected by stratospheric temperature or by stratospheric adjustment processes. ...

These ERA5LBL (all-sky, clear-sky, and dry) and idealized experiments together clarify the **robustness, physical basis, and conditions** of the linear relationship. The seasonal and spatial variations simulated by ERA5LBL also mark a large range of climate states that Earth condition is unlikely to exceed (L184). A more comprehensive theoretical explanation remains an open question, which we intend to address in a full-length study. In particular, future work will explore the following key questions:

- Why the linear relationship appears in monochromatic wavenumbers.
- Why the linearity is enhanced by CO₂ spectrum – the work by [Jeevanjee et al., 2021] that idealize the absorption spectrum of CO₂ could be used to explain the linearity of CO₂ forcing in dry atmosphere.
- Why the OLR-IRF relationship is so hard to break down by clouds and why the spectroscopy of WMGHG further enhances this phenomena.
- What determines the regression intercept, which marks the important transition from negative to positive forcing, and is this transition physically meaningful for planetary climate.

These questions arise from LBL radiative transfer and its idealization, for which a complete analytical framework to address them could hardly be fully conveyed in this short article. An eventual theoretical formulation will likely idealize the detailed physics represented by the LBL results. **The LBL results therefore defines the physical relationship that future analytical theory would aim to reproduce and explain**, rather than further support.

We also added the following statements to link to the “emergent constraint”:

Although line-by-line calculations under all-sky conditions could, in principle, benchmark LW IRF for each model, they are computationally prohibitive and cannot fully represent model-specific cloud–radiation interactions. Our regression framework provides an efficient alternative, yielding accurate LW IRF estimates across diverse atmospheric and cloud states characterized by each model’s OLR. This implication to ESMs is conceptually similar to an emergent constraint [Hall et al., 2019].

3. *The manuscript covers two somewhat disparate topics: a) the temporal evolution from 2020-2025 of instantaneous radiative forcing (IRF) by CO₂, CH₄, N₂O, and two sets of halocarbons and b) the role of parameterization error in explaining the range of IRF across a subset of models contributing to CMIP. These are somewhat interwoven in that Figure 3 and the relevant text, on the latter subject, comes between figure 1-2 and figure 4, which address the former. The confusion also arises in the introduction, where the material describing the precision with which real-world IRF is known veers into a discussion of variability across earth system models at line 61. These are distinct subject, linked here only by the use of the same regression technique to explore them. The manuscript would benefit from making the conceptual distinction between these questions more clear and not, for example, equating diversity across ESMs with uncertainty (line 62). It is not clear that the two subjects belong in the same manuscript.*

We thank the reviewer for this helpful comment. In context (b), we used ‘uncertainty’ to denote the inter-model spread in CMIP simulations, following the IPCC convention where ERF uncertainty is reported as model diversity because ERF is not directly observable and depends on parameterizations, cloud states, and adjustment processes. We agree that the original manuscript used the term uncertainty inconsistently between the two contexts, which could cause confusion.

We have revised the abstract, introduction (L68-76), L288-307, and the conclusion, using the term ‘spread’ or ‘discrepancy’ instead of ‘uncertainty’ when discuss ESMs. Making this distinction indeed help to clarify the message that the community can use the regression method to benchmark model-simulated IRFs and to remove biases from each model, such that a better agreement in the effective radiative forcing for future climate assessments can be reached.

4. *More detailed comments: Errors in “humidity and cloud fields in reanalyses and ESMs” (line 60) have both very different causes and different implications for computing IRF. On the one hand the IRF to which a model is subject depends only on the model state - the uncertainty (line 62) arises only from errors in approximation. Reanalyses might well be biased in ways that affect OLR and forcing calculations but the cited literature does not support this claim - and indeed the authors are happy to make calculations based on ERA5. Should these not be trusted after all?*

Thanks for this comment, we agree completely with the comment; the fact that the **IRF computed from ERA5 could not be fully trusted (L65, L132-135) is exactly the reason why we need the regression-based method to constraints IRF from observed OLR, rather than using ERA5LBL calculation directly.**

On the other hand, we hope this issue has been well addressed by the regression method constrained by ERA5LBL (L180-191) because:

- 1) A wide range of gridded, monthly-mean **clear-sky** conditions are used to construct the regression coefficients and is well represented by it. As long as the global-mean clear-sky state of Earth does not exceed the range, the results are not affected by biases in ERA5 (L182,L476-479).
 - 2) The all-sky results of ERA5LBL are used to **evaluate, not to construct**, the linear coefficients. Thus when the observed OLR is applied to the regression model, any biases in ERA5 cloud state or the cloud-radiation parameterization do not affect the results.
5. *The paragraph stating on line 68 does set the stage for the work that follows. It seems possible that the authors want to anticipate applying their regression results to CERES observations of broadband irradiance; this might be made more clear by omitting some of the more general description.*

Thanks, rephrased, see L80. This paragraph was also intended to link to existing studies on attributing greenhouse effects from satellite observations.

6. *The assertion on lines 77-79 is more dramatic than warranted. The uncertainty in estimates of clear-sky radiation is of order 1%, as [14] notes; applying the GRT code*

or any line-by-line model to greenhouse gas concentrations from 2000-2024 using the small number of samples from that paper would be a much simpler exercise than the present manuscript undertakes. If the statement is meant to apply to clouds the authors might be more precise.

The small uncertainty about 1% in [14] only accounts for uncertainty in spectroscopy and radiative transfer process at **prescribed atmospheric states** (ERA5 clear-sky condition at 2014), but does not have any skill to evaluate the forcing caused by biases or uncertainty in atmospheric states (**state-dependency**). Because clear-sky and cloud effects must compensate for each other (as the simulated all-sky TOA fluxes are tuned/assimilated towards observation), uncertainties in OLR caused by that in clear-sky and clouds should be of similar magnitude.

7. *How are clouds treated in the ERA5LBL computations?*

Cloud treatments are described in L353-361. We also state in L362-372 that when altering cloud-radiation parameterizations, all-sky LW IRF computed from ERA5LBL can vary by 10% for results presented in Figure 1, but does not break down the good agreement with clear-sky derived linear regression model, so that it has no effect on the constrained IRF once CERES OLR is used in Figure 2.

8. *The value of IRF as measured at the tropopause results in part from the strong adjustment of stratospheric temperature to the increases in CO2 that constitute the majority of 20th and 31st—century forcing. Are the authors aware of evidence that this also holds for other greenhouse gases for which the stratospheric adjustment is small?*

Some discussions have been added to L452-465.

To answer the question specifically, the tropopause level is conventionally used to evaluate IRF due to CO2 and ozone. For other WMGHGs, the difference between TOA and tropopause IRF is small, and the adjusted forcing, in theory, must lie within the range of TOA and tropopause IRF. Evaluating the IRF from either TOA or tropopause makes little differences (table below). For ozone (not discussed in this article), the adjusted forcing is also much closer to the tropopause value than the TOA. We did not evaluate stratospheric h2o in the table though.

Forcing agent	TOA	Tropopause	Adjusted
PI greenhouse gas	2.748	3.852	3.708
PI CO2	1.373	2.292	2.146
PI CH4	0.616	0.525	0.582
PI N2O	0.181	0.185	0.189
PI O3	0.080	0.462	0.352
PI HCs	0.415	0.307	0.360

Table 1: IRF at TOA, IRF at tropopause, and adjusted forcing, computed using RTE-RRTMGP following RFMIP experiment [Pincus et al., 2020]. Forcing is evaluated with respect to present-day climate.

9. *Lines 114-121 are not consistent with my reading of [14]. Their Table 3 shows a range of all-sky IRF/clear-sky IRF at the top of atmosphere across models as being something like 0.735-0.767. The value reported for LBLERA is 0.734 - which seems consistent, not inconsistent, with table 3.*

The reported value for ERA5LBL was **0.80 (2.16 wm-2 (L130) /2.70 wm-2)**, indeed higher than the range 0.735-0.767 reported by Table 3 of [Pincus et al., 2020]. The 0.734 referred by reviewer here is likely the ratio between clear-sky TOA and tropopause IRF?

The higher value suggests ERA5LBL has weaker longwave cloud masking effect than the three CMIP6 models. To be more specific, ERA5LBL simulates 21 wm-2 of LW cloud effects, but the LW cloud radiative effects of GFDL's AM4 is 23 Wm-2, IPSL-CM6A-LR is 25 Wm-2, and HadGEM3-GC31-L is 22.5 Wm-2 (three models used in Table 3 of [Pincus et al., 2020], IPSL has lowest all-sky IRF/clear-sky IRF and HadGEM3 has higher values), all higher than ERA5LBL.

Although the cloud effect of ERA5LBL is not the truth, the combined results with clear-sky (contain compensating biases) yield excellent agreement with all-sky OLR from CERES EBAF v4.2. This good agreement can not be reached when combine clear-sky conditions of ERA5, but cloud conditions simulated by other models, not even the actual cloud effect if it can be truly retrieved.

Some relevant discussions can be found in reply to Reviewer 3' comment 3 .

10. *Lines 122-128 seem irrelevant. The simplified expressions for radiative forcing described here are not intended as benchmarks in the same sense as [14] or the present work.*

Thanks for the comment, it indeed makes sense in writing to remove the discussion here to make the main text more compact. We have moved these comparisons to Method (L512-544).

11. *Line 129-130: Clouds are not constrained by any observations in ERA5.*

Rephrased. Clouds are not directly assimilated but it is indirectly improved compared to previous reanalysis products as ERA5 for the first time assimilate all-sky microwave radiances (helps improve humidity, temperature, and surface). Nevertheless, this sentence was meant to convey that the direct result from ERA5LBL can not be fully trusted despite the improvements.

12. *The meaning of lines 151-152 is not clear.*

Rephrased to improve the clarity:

The regression intercept b is obtained from Extended Data Figure 3, and the slope a is then determined as $a = (\bar{R} - b)/\bar{F}$, where \bar{R} and \bar{F} denote the global-mean clear-sky OLR and IRF in 2010. Under this definition, r is 0 by construction for clear-sky conditions.

13. *Line 158 reports that regression coefficients between OLR and forcing derived for clear skies hold equally well for profiles including clouds. Is this true for some fundamental reason, or are the errors incurred in the translation just small because forcing is a small fraction of OLR?*

Great question. The similarity in linearity when clouds are added is likely to be non-numeric. For example, the maximum longwave cloud effects in Figure 1(c) and Extended Data Figure 3 are over 100 Wm^{-2} , leading to a difference of 5 Wm^{-2} in LW IRF at the tropopause of $4x\text{CO}_2$. This difference is explained by the regression slope $a = 0.051$. This phenomenon is summarized more physically around L178 as:

Conditions for this linear relationship to hold are further discussed in the Methods section and in Extended Data Figure 5. These statistics confirm that regions with similar OLR, regardless of atmospheric or cloud conditions, exhibit nearly identical LW IRF.

The conserved linear regression slope by clouds seems to hold at every monochromatic wavenumber, as we found in Extended Data Figure 5(d) that, even when the added overlapping spectrum is random noise at a random tropospheric layer, the linearity still holds. More detailed studies on analytical explanation will be conducted in the future.

14. *That the regression methods can reconstruct forcing from observations of OLR is a neat trick. It is less clear what differences between LBLERA, the regression model applied to LBLERA (which agrees by construction), and regression applied to observations tells us about the IRF to which the world is subject, especially given that differences are of order 0.025 W/m^2 out of 9 W/m^2 .*

We thank the reviewer for raising this important point. We have now significantly expanded the discussion on uncertainty and applications in L466-520. Scripts are provided to apply the method and evaluate uncertainties from user-specified OLR uncertainties.

For $4x\text{CO}_2$, the uncertainties include 0.196 Wm^{-2} from gas spectroscopy, 0.10 Wm^{-2} from the regression method, and 0.094 Wm^{-2} from observation; they combine to 0.24 Wm^{-2} .

Additionally, we believe that the regression models have been tested and evaluated **independently**. In L207, we improved the clarity on the message that the regression is trained on clear-sky ERA-LBL for 2010, but then applied out-of-sample to all-sky ERA-LBL across multiple years.

In addition, we have applied the regression model to

- 1) clear-sky IRF obtained from four other independent LBL codes (Table S1);
- 2) to clear-sky RRTMGP calculations (bias removed based on independent evaluation reported for RFMIP) for GCM simulations (Figure 3(b) and Extended Data Figure 8 for amip, piclim-control, piclim-4xco2; their atmospheric condition is quite different). L275-287 has been rephrased to make this evaluation clearer.

Figure 1: A comparison between GCM all-sky double-call and regression estimate of LW IRF at tropopause (Wm^{-2}). Dashed curves are from fixed-SST atmosphere-only simulation, and solid curves are coupled abrupt-4xCO₂ experiment (SST warms up with time). Black curves are all-sky double-call results computed by RTE-RRTMGP in GFDL’s AM5/CM5 prototype with 0.33 Wm^{-2} bias removed, and gray curves are regression estimate (Eq. 3) using model simulated-OLR. The online model’s cloud conditions (different convection and microphysics scheme) and cloud-radiation parameters (two-moment scheme resolve effective radius; band-by-band optics; longwave-scattering approximation) are different from ERA5LBL, yet the agreement is good across evolving model states.

3) and to all-sky double call performed in GFDL’s AM5/CM5 prototype (Figure 1 in this response). This experiment is not added to the main text because the model has not been published and the results are expected from Figure 4. This comparison is reassuring because CM5’s cloud conditions are quite different from ERA5 due to different microphysics and convection parameterizations, the cloud-radiation interaction is also configured differently.

In these applications, the deviation in regression estimates is well within the estimated regression uncertainty. For clouds, we have also elucidated in comments 4, 7, and 13 that cloud treatments in ERA5LBL simulations changes their forcing and does not affect the agreement with the regression model.

Responses to Reviewer 2

1. *Summary: The manuscript “A Strong Constraint on Radiative Forcing of Well-mixed Greenhouse Gases” by Feng et al. provides the first benchmark calculation of the long-*

wave instantaneous all-sky forcing of well-mixed greenhouse gases (WMGHG), both at the top of the atmosphere and the tropopause. To achieve this, they use a computationally optimized but very accurate line-by-line radiative transfer model based on atmospheric input from reanalysis data over a period of more than 20 years. The authors use this benchmark to derive an empirical regression model utilizing the strong relationship between outgoing longwave radiation (OLR) and radiative forcing that is remarkably similar between clear-sky and all-sky conditions. This regression model in turn is applied to observed OLR for both clear-sky and all-sky conditions, yielding very good agreement with the line-by-line simulations. This way, they incorporate the effects of clouds and changing temperatures on the forcing and provide a useful tool to diagnose instantaneous radiative forcing from coupled climate models. The study addresses a central and relevant question in climate science: Despite a generally very good understanding of atmospheric radiative transfer, uncertainties in WMGHG forcing have been persistent due to limitations of radiation parameterizations, biases in the atmospheric states of climate models, as well as the effects of clouds and of changing global temperatures. Whereas previous studies on this topic have addressed some of these challenges, a major advance in this work is that they simultaneously address all these challenges. By combining the advantages of line-by-line radiative transfer simulations, reanalysis data, and satellite observations, this work better constrains the magnitude of the longwave instantaneous all-sky forcing of the most important greenhouse gases. The manuscript is clear and well-written. The authors use a highly accurate radiative transfer code, combined with a state-of-the-art reanalysis dataset. They derive a regression model that is then applied to a well-established observational satellite product, which is used to demonstrate the robustness of their approach. They advance the field by providing a better constraint on the longwave instantaneous all-sky WMGHG forcing and a tool to diagnose it from Earth system models. Their findings are highly relevant for the atmospheric radiative transfer community but a better constraint on the radiative forcing of WMGHG also has broader implications for climate science more generally. In my opinion, the manuscript is therefore suited for publication in *Nature*. I enjoyed reading this manuscript and only have some minor comments.

Thank you for a concise and comprehensive summary of this article, we appreciate the timely and encouraging comments.

2. While the authors provide good justifications to focus on the longwave forcing component (much larger magnitude and higher uncertainty), the manuscript would be even stronger if it also discussed implications (or possible extensions) of this work for the shortwave component. Building on the strong relationship between outgoing longwave radiation and longwave forcing, one could, for example, imagine a similar relationship between reflected shortwave radiation and the shortwave forcing. Have the authors tested this? Or is there a reason why it was not feasible in this setup? If so, what would be necessary to enable such an extension to the shortwave domain? A discussion of this would make the manuscript even more useful to the community.

Thanks, it is a good point about the shortwave component of forcing. We have tested this idea when preparing the manuscript. We found that there is indeed a very good

linear relationship between clear-sky SW IRF and TOA SW fluxes (surface albedo has scalable impacts on IRF and SW UP at TOA), but this good relationship **break down by clouds**. Clouds could both mask (reduce path length) and enhance (reflection) IRF, such effects are difficult to separate using the TOA fluxes alone. An alternative formulation to constrain on shortwave IRF is required, for which I think the model needs to differentiate masking and reflection of clouds efficiently using observable quantities (maybe by combining observed shortwave fluxes at the surface).

Considering shortwave component accounts for less than 10% of IRF, and that constraining the LW component has greatly reduced the spread in models, we chose to leave the shortwave component as an open question to be addressed by community (L298-307).

3. *The authors mainly analyze the forcing at the tropopause. While this is made clear throughout the manuscript, it is not clear from the abstract and conclusion, where it is simply referred to as “instantaneous forcing”. In my opinion, this undersells the achievements of this study, because the tropopause forcing is more closely linked to the effective forcing after stratospheric adjustments and thus the more relevant quantity for surface warming compared with the instantaneous forcing at the top of the atmosphere. I suggest also mentioning this point in the conclusions.*

Thank you for this important suggestion. Following this comment, we have improved the abstract (L24-26), L237-238, and the conclusion (L316),

4. *The regridding of the ERA5 data to a much coarser resolution (around 250km rather than the native 31km) is understandable given the expensive nature of line-by-line radiative transfer. However, it should not just be mentioned in passing in the methods but rather be made more transparent by mentioning it in the main text and briefly discussing the limitations of this approximation, particularly on cloud effects (the latter can be done in the methods).*

Thanks for pointing it out, in our HPC server, it is technically very challenging to conduct full resolution calculations routinely due to huge memory required to process the data.

Following the suggestion, we conduct ERA5LBL calculations at 2.5 x 2.5 degree resolution (144x90 gridcell) for year 2010. This results in differences in present-day forcing of less than 0.01 Wm^{-2} . It makes 4xCO2 forcing 0.01 Wm^{-2} larger, causing slightly more visible differences in Figure 4 between the regression-based estimate and ERA5LBL from coarser resolution, but are very small compared to the uncertainty range.

Information on the resolution has been added to Methods:

To minimize computational costs, we conduct radiative transfer calculations at a resolution of **2.5° for the year 2010, which is used to construct the regression model**, and at a coarser resolution of **7.5 ° for other years that are used to evaluate the model**.

5. *Specific comments:Line 55-56: The sentence should start with an upper-case “L”.*

Revised.

6. *Line 72: It should either say "often exceeding" or "that often exceed".*

Revised.

7. *Lines 167-169: Please provide more details on how exactly the confidence range is estimated from the meridional means (either here or in the methods).*

Following the suggestions, more details have been added to here:

The 5–95% confidence range, estimated from monthly LW IRFs across 7.5° longitude bins over 24 years (Extended Data Figures. 6-7), is 0.064 W m^{-2} ($\sim 2\%$), based on ± 1.96 times the standard error in the regression estimates.

We have also added a ‘Evaluation and Uncertainty’ to Methods (L466-498).

8. *Lines 179-180: The part of the sentence after the semicolon is presumably missing a verb.*

Revised.

9. *Line 246: Does this refer to the method described in lines 167-169? Please clarify.*

Yes. This section has been majorly reorganized to improve the clarity, and the formulation on 4xCO₂ forcing and PD CO₂ forcing has been combined.

10. *Lines 250-252: This sentence is confusing to me, as it appears to imply that the forcing increase is caused by the surface warming (i.e., the response). Please rephrase.*

We thank the reviewer for pointing this out. When OLR is anomalously high, as it has been in recent years due to surface warming and weakened cloud LW effects, the diagnosed IRF also appears high. This result implies that 4×CO₂ forcing strengthens in a warmer, less cloudy climate, offsetting approximately 5% of the longwave feedback, and about 10% of the longwave+shortwave feedback.

This claim is further supported by the IRF analyzed from the radiation double-call in an abrupt 4xCO₂ experiment using a prototype of GFDL’s CM5 (Figure 1 of this response), although we did not include the result in the main text because the model has not been published. This enhanced forcing component is simulated by GCMs but is accounted for as a residual term when apply the radiative kernel in the forcing-feedback analyze framework, or as ‘feedback’ when adopt ‘Gregory method’ [Gregory et al., 2004]. We plan to address it collaboratively in future studies.

11. *Line 269: The reference should presumably be to Fig. 3c?*

Yes, thank you, corrected.

12. *Line 306: An “and” is missing after the last comma.*

Revised.

13. *Line 325: “annual-mean, monthly-mean”: does this mean that both are used? Please rephrase.*

Thank you, deleted the extra 'annual-mean'.

14. *Fig. 2: Please explain the error bar in the legend and caption.*

Thank you. Updated.

15. *Fig. 3: Please explain the meaning of the colored shadings in the legend and caption.*

Revised.

16. *Remarks on code availability: The provided code is generally well-documented, including a README file describing how to build GRTcode and a Guide file outlining the steps to run GRTcode for the reanalysis input. As far as I can tell, there does not seem to be any documentation how the data is post-processed to produce the figures in the manuscript.*

Thank you for pointing out! Scripts for generating key figures based on the broadband data have been updated, they produce the essential data, coefficients, and for evaluation of uncertainties. The outputs are summarized into a spreadsheet (uploaded to Zenodo) to visualize Figure 1(a) and Figure 2. Scripts for implementing the regression model are also uploaded and can be tested in Code Ocean.

Responses to Reviewer 3

1. *This paper seeks to definitively establish benchmark calculations of the radiative forcing of greenhouse gases over the last several decades. It does so by performing a massive number of radiative transfer calculations with realistic atmospheric states as they occurred in over the last several decades, so that the calculations are fully aware of the atmospheric state and how that impacts these calculations. The paper represents an impressive result technically because of the sheer volume of radiative transfer calculations necessary to achieve this paper’s results. A vast amount of computational resources were expended for this paper in order to create the definitive benchmark of greenhouse gas radiative forcing calculations. It may be acceptable for publication with major revisions.*

We thank the reviewer for this encouraging summary of our study.

There are several revision needs. First, it is that it is unclear if the impressive technical results fundamentally alter the scientific community’s understanding of the radiative forcing of greenhouse gases, or if the calculations are just about reducing uncertainty in the quantity. Specifically, do they show that the community needs to keep doing this this level of calculation moving forward?

Our intention was not to suggest that such extensive LBL calculations should continue to be performed routinely. Rather, the purpose of this work is to use these comprehensive benchmarks to develop a simple regression framework that can reproduce LBL-calculated LW IRF with high accuracy under varying atmospheric states. This regression model provides a computationally efficient alternative to direct LBL

calculations while explicitly accounting for the dependence of radiative forcing on temperature, humidity, and cloud conditions.

Following the reviewer’s suggestion, we have clarified this point and strengthened the application aspect of the study:

- We refined the regression equation and validated it across a wide concentration range from PI to four times present-day WMGHG. This required additional simulations at intermediate perturbation levels and the introduction of a second-order polynomial to describe the concentration dependence of the regression slope (Tables 1-2; L157, L404-412).
- To emphasize that future users need not repeat the large-scale LBL calculations, we added an “Application” subsection in the Methods (L500-520). This section provides scripts and instructions for deriving LW IRF for any well-mixed greenhouse gas using only its concentration and OLR as inputs. The narrow uncertainty range reported in this study remains valid for concentrations within the PI to 4x2010 level range (L510).
- Outside the range that could possibly be examined by global-scale LBL, it is recommended to continue the sampling strategy conducted by existing studies for constructing the simple formulas, but with additional scaling/constrained based on the OLR-regression method, as discussed in L532-544:

The robust linear relationship between OLR and LW IRF presented in this study provides a simple rule of thumb: accurate LW IRF estimates can be obtained from limited sampling if the weighted-mean OLR of the sampled atmospheric and cloud states matches the observed global mean. This finding suggests that biases in existing simplified formulas likely arise from lower mean OLR in the sampled conditions rather than from the functional form of the formulas themselves. Limited sampling is particularly useful because it enables computationally efficient calculations across a wide range of concentrations compared with global-scale LBL calculation. The OLR–regression model (Eq. 2) can therefore be applied as an additional constraint when sampling the observed climate [Myhre et al., 2006] or used to rescale forcing estimates from the sampled mean OLR to the OLR of a target climate scenario. Incorporating this approach enables more reliable estimates for extreme WMGHG perturbations across diverse climate conditions.

We hope these changes would clarify that the goal of this work is to reduce, not require, the computational burden of LBL calculations by offering a simple regression approach that also allows observational constraints and flexible integration into other studies.

2. *In order for the manuscript to rise to be of broad interest, the authors need to discuss in detail if this comprehensive result alters our understanding of radiative forcing of these gases in general, and if it alters how the community should consider what is sufficient for estimating greenhouse gas radiative forcing. The authors do compare their epic*

calculations to established GHG RF formulae and it appears that their result does alter and upward-revise the understanding of greenhouse gas RF based on a small number of calculations, e.g., in Etminan et al, 2016 and Pincus et al, 2020? Why is this the case? Does a more complete sampling of the atmosphere need be included in the formulae? Were the impacts of clouds underestimated in previous simple formulae and can they be represented in formulae moving forward? Is an underestimation guaranteed unless there are large-scale calculations such as those presented here? Or can better sampling of atmospheric states for formulae generation be done? Given the calculations that the authors performed, a quick analysis of the data already in hand would answer this question and I recommend the authors do that analysis.

We thank the reviewer for this insightful and comprehensive comment. We have expanded the discussion in L500-520 to clarify the broader implications of our results.

In direct response to the reviewer’s questions:

- (a) Why upward-revise: [Pincus et al., 2020] produces lower all-sky IRF because CMIP6 models used in their results have stronger cloud effects than ERA5LBL. When combine ERA5 clear-sky condition (used in Pincus et al., 2020) with CMIP6 model cloud effects, the OLR is not expected to be consistent with observed OLR. This issue is also discussed in response to Reviewer 1’s Comment 9.

For [Etminan et al., 2016], discrepancies may stem from both sampling biases (temperature, humidity, and clouds) and cloud–radiation parameterizations (e.g., optics, overlap) that may not be identical to how the observational cloud products are derived. These differences are expected to minimize if the sampling/weighting follows the OLR constraint as recommended above.

- (b) Sampling and need for large-scale calculations: The differences between the simple formulae and our benchmark are not a direct consequence of subsampling. They arise from the mismatch between the mean OLR of the sampled states and the observed global mean. Our analysis suggests that if sampling profiles are constructed such that their weighted-mean OLR matches observations, the resulting IRF is expected to agree well with the ERA5LBL or the regression estimate.

3. *The reason it is important to connect these results back to simple formulae for each gas species is that there are large communities that use such formulae for, to name just a few examples, adaptation, mitigation, and resilience planning. If a simple formula is insufficient because it is low-biased currently, or because it will always be low-biased due to its simplicity, then many communities need to know about it and know that achieving accurate GHG RF estimates for applications purposes is not straightforward. Alternatively, can the community have confidence that a simple formula or formulae is sufficient, perhaps with tweaking from calculations like these. Or can the authors say that the level of calculation they performed needs to be done once but not again? Or is that problematic because the atmospheric state, including thermodynamics and clouds (and its response to forcing), will continue to dominate the RF error budget? Or is spectroscopy now and into the future the driver of uncertainty?*

There is much that can be gained in understanding from such an epic set of calculations, and the paper needs revisions, including an assessment of whether tweaks to simple formulae will be sufficient for assessing greenhouse gas RF moving forward, especially for individual gases.

We thank the reviewer for raising these broad and important questions. To address the comment, we have added specifically an Application subsection in *Methods* (L499-520) to explain how the regression-based method can be applied by broader community.

To address the reviewer's specific questions:

- (a) Are simple formulae inherently low-biased?

No, the simple formula is low-biased likely due to the mean OLR of their samples is lower than CERES EBAF or ERA5LBL. Their results can be rescaled using the OLR-regression model proposed in this study, as addressed in the next question.

- (b) Will this level of LBL calculation be required again?

No. The regression model derived from our massive calculations, is as simple to use as existing formulas, and additionally accounts for state-dependency. As long as the background OLR remains within Earth-like limits, future assessments for well-mixed GHGs (vertically uniform, within PI to 4x2010 level range) can rely on the regression method without repeating full-scale LBL computations.

For WMGHG perturbations outside the range, we recommend simple formulas to be further performed but rescaled using (OLR-b) following Eq. 2 and Table 1 (b is the regression intercept, fixed for each WMGHG), such that they have the ability to account for variations in temperature, humidity, and clouds. This rescaling can be very important for paleoclimate or extreme future climates (if we believe the global-mean is within the range of polar winter to tropical summer of present-day Earth) that has distinct climate states from the samples used to build simple formulas (L542).

- (c) The error budget in IRF

With the OLR-regression constraint, the residual uncertainties in greenhouse gas RF primarily arise from three sources: (1) regression uncertainty, associated with how variability in thermodynamic and cloud fields leads to proportional variations in OLR and IRF (L181-191), (2) spectroscopic uncertainty, evaluated as discrepancies across five LBL models, and (3) OLR observational uncertainty.

These sources are discussed more explicitly in L467-498. The **regression uncertainty is comparable to the spectroscopic uncertainty for most gases**, except for CO₂ and CFCs/HFCs, where spectroscopy contributes more. Uncertainties from OLR observations, when propagated to forcing, are generally smaller than those from regression or spectroscopy.

- (d) The error budget in adjusted forcing We believe that remaining uncertainties come from **shortwave** IRF component and **tropospheric adjustment**, particularly the cloud adjustment processes. A more explicit discussion has been added:

The remaining spread (yellow shading in Fig. 3c) arises from discrepancies in shortwave IRF ($-0.52 \pm 0.48 \text{ Wm}^{-2}$) and adjustment processes ($-0.20 \pm 0.38 \text{ Wm}^{-2}$). Unlike the longwave component, a simple relationship between TOA fluxes and shortwave IRF is less likely to hold, as surface albedo primarily controls reflection, whereas clouds not only reflect but also mask portions of the shortwave IRF, effects that TOA fluxes alone may not fully distinguish. Alternative formulations [Roemer et al., 2025, Goodwin et al., 2025], along with improved TOA [Gristey, 2025] and surface [Shi et al., 2025] shortwave observations, may help to better constrain these uncertainties. Because the adjustment process is not independent of radiation [Byrom and Shine, 2022, Shine et al., 2022, He et al., 2023], further reductions in inter-model spread could be achieved if consistent radiation parameterizations were adopted.

Using CMIP6 data, we found that the remaining spread is more statistically correlated with the shortwave IRF (tropopause) component. But when look into GFDL’s model that uses the same shortwave code, we found the ERF is quite sensitive to convective and cloud parameterizations that lead to substantial differences in cloud adjustment process. Our framework allows this decomposition (LW IRF tropopause, SW IRF tropopause, adjustment) to be straightforwardly conducted in ESMS via a double-call method, which we plan to use for more detailed future investigations.

4. *Finally, I’ll comment on the point the make authors showing that LW IRF is related to clear-sky OLR. This is an intriguing result. From this, the take-away message is that as long as the remote sensing community continues to accurately measure this quantity with satellite instrumentation, then the larger scientific community can use their linear regression to derive a metric of LW IRF for all greenhouse gases. Are the reported accuracies and precision values of satellite-derived OLR products sufficient to do this and are their implications for the long-term on-orbit stability that instruments and associated OLR products need to achieve?*

We thank the reviewer for this excellent summary of take-home message.

The all-sky LW IRF is determined directly from the observed all-sky OLR, while the regression coefficients are derived from clear-sky OLR. Substantial clarifications on this relationship have been added to the *Methods*. In this study, we report an uncertainty range derived from the 2.5 Wm^{-2} global-mean uncertainty of OLR in the CERES EBAF v4.2 product, but we do not explicitly quantify the long-term on-orbit stability of satellite observations. When the regression framework is applied with alternative OLR products, their associated uncertainties can be accounted for following **L484**. This has also been implemented in the shared scripts, allowing users to readily adapt the method for different datasets and applications.

Nevertheless, we note that the observed increase in global-mean OLR over the past 24 years is supported by multiple independent lines of evidence, including CERES observations, ERA5 reanalysis, sea surface temperature, and surface radiation records, providing confidence in the long-term stability of the OLR record. Should future

assessments identify a systematic drift in an OLR product (e.g., $x \text{ W m}^{-2} \text{ decade}^{-1}$), the corresponding drift in the inferred IRF can be estimated as $a \times x$, where a is the regression slope in Eq. (2). For present-day CO_2 , $a \approx 0.01$, indicating that the IRF drift is only about 1% of the OLR drift.

We have also revised the main text accordingly to emphasize the role of long-term observational stability:

Confidence in this benchmark is reinforced by a regression method that integrates observational constraints from satellite-observed OLR to account for uncertainties from cloud effects and evolving climate states. In the future, such tight constraints for the LW IRF are achievable only with the continuation of stable, long-term satellite records of Earth’s energy fluxes [Loeb et al., 2024].

and in the *Methods*:

The observational uncertainty, σ_o , should be recalculated for the specific OLR product used. Although long-term instrumental stability is not explicitly treated here, any known drift in the OLR record can be propagated to IRF by scaling it through the regression slope a . Because a is small (e.g., 0.0512 W m^{-2} for $4 \times \text{CO}_2$), the resulting IRF drift is expected to be much smaller than that of the OLR record itself.

5. *A brief review of the code indicates that it is thorough and usable for the community. I did not install or run the code, but it does have a README file.*

Thanks, following suggestions from the review, we have added example scripts to implement the regression model for IRF estimates directly from OLR and gas concentrations. The scripts are expected to be adapted by the community easily.

References

- [Byrom and Shine, 2022] Byrom, R. E. and Shine, K. P. (2022). Methane’s solar radiative forcing. *Geophysical Research Letters*, 49(15):e2022GL098270.
- [Chen et al., 2024] Chen, Y.-T., Merlis, T. M., and Huang, Y. (2024). The cause of negative CO_2 forcing at the top-of-atmosphere: The role of stratospheric versus tropospheric temperature inversions. *Geophysical Research Letters*, 51(1):e2023GL106433.
- [Cronin and Dutta, 2023] Cronin, T. W. and Dutta, I. (2023). How well do we understand the planck feedback? *Journal of Advances in Modeling Earth Systems*, 15(7):e2023MS003729.
- [Etminan et al., 2016] Etminan, M., Myhre, G., Highwood, E. J., and Shine, K. P. (2016). Radiative forcing of carbon dioxide, methane, and nitrous oxide: A significant revision of the methane radiative forcing. *Geophysical Research Letters*, 43(24):12–614.

- [Goodwin et al., 2025] Goodwin, P., Williams, R. G., Ceppi, P., and Cael, B. (2025). Climate feedbacks derived from spatial gradients in recent climatology. *Journal of Geophysical Research: Atmospheres*, 130(12):e2024JD043186.
- [Gregory et al., 2004] Gregory, J. M., Ingram, W. J., Palmer, M., Jones, G. S., Stott, P., Thorpe, R., Lowe, J. A., Johns, T., and Williams, K. (2004). A new method for diagnosing radiative forcing and climate sensitivity. *Geophysical research letters*, 31(3).
- [Gristey, 2025] Gristey, J. J. (2025). A perspective on shortwave radiative energy flows in the earth system. *Advances in Atmospheric Sciences*, pages 1–12.
- [Hall et al., 2019] Hall, A., Cox, P., Huntingford, C., and Klein, S. (2019). Progressing emergent constraints on future climate change. *Nature Climate Change*, 9(4):269–278.
- [He et al., 2023] He, H., Kramer, R. J., Soden, B. J., and Jeevanjee, N. (2023). State dependence of co2 forcing and its implications for climate sensitivity. *Science*, 382(6674):1051–1056.
- [Jeevanjee et al., 2021] Jeevanjee, N., Seeley, J. T., Paynter, D., and Fueglistaler, S. (2021). An analytical model for spatially varying clear-sky co 2 forcing. *Journal of Climate*, 34(23):9463–9480.
- [Loeb et al., 2024] Loeb, N. G., Doelling, D. R., Kato, S., Su, W., Mlynchak, P. E., and Wilkins, J. C. (2024). Continuity in top-of-atmosphere earth radiation budget observations. *Journal of Climate*, 37(23):6093–6108.
- [Myhre et al., 2006] Myhre, G., Stordal, F., Gausemel, I., Nielsen, C. J., and Mahieu, E. (2006). Line-by-line calculations of thermal infrared radiation representative for global condition: Cfc-12 as an example. *Journal of Quantitative Spectroscopy and Radiative Transfer*, 97(3):317–331.
- [Pincus et al., 2020] Pincus, R., Buehler, S. A., Brath, M., Crevoisier, C., Jamil, O., Franklin Evans, K., Manners, J., Menzel, R. L., Mlawer, E. J., Paynter, D., et al. (2020). Benchmark calculations of radiative forcing by greenhouse gases. *Journal of Geophysical Research: Atmospheres*, 125(23):e2020JD033483.
- [Roemer et al., 2025] Roemer, F. E., Buehler, S. A., and Menang, K. P. (2025). How to think about the clear-sky shortwave water vapor feedback. *npj Climate and Atmospheric Science*, 8(1):1–10.
- [Shi et al., 2025] Shi, C., Letu, H., Nakajima, T. Y., Nakajima, T., Wei, L., Xu, R., Lu, F., Riedi, J., Ichii, K., Zeng, J., et al. (2025). Near-global monitoring of surface solar radiation through the construction of a geostationary satellite network observation system. *The Innovation*, 6(5).
- [Shine et al., 2022] Shine, K. P., Byrom, R. E., and Checa-Garcia, R. (2022). Separating the shortwave and longwave components of greenhouse gas radiative forcing. *Atmospheric Science Letters*, 23(10):e1116.

[Young et al., 2011] Young, P. J., Thompson, D. W., Rosenlof, K. H., Solomon, S., and Lamarque, J.-F. (2011). The seasonal cycle and interannual variability in stratospheric temperatures and links to the brewer–dobson circulation: An analysis of msu and ssu data. *Journal of Climate*, 24(23):6243–6258.

Response to Reviewer Comments

Referee 1

- *The revision has clarified some points but the manuscript structure remains essentially unchanged, so that the work represents (as noted by reviewer 3) a new approach for computing radiative forcing based on outgoing longwave radiation without any change in understanding or conception. The approach might provide a higher-accuracy estimate of radiative forcing than does computation with a finite number of profiles [13, 14, 23], especially with respect to cloudy atmospheres, and so might have value in the specialist literature. It’s hard to see how modest improvements in the accuracy of an already well-constrained problem, based on massive computation but absent physical arguments, warrant publication in a high-impact journal even in the absence of confusion and conflicts with previous work as outlined below.*

We thank the referee for this perspective. While the radiative transfer process itself is indeed well understood and line-by-line calculations are highly accurate, **radiative forcing for the observed climate is not well constrained** because it depends on the realized distributions of temperature, humidity, and clouds. Because the atmospheric state is complex and chaotic, the global, layer-by-layer profile cannot be observed perfectly; the reliability of forcing estimates for the present-day climate has remained difficult to quantify, as discussed in **L57–83**.

This work demonstrates that radiative forcing can be robustly constrained under realistic and evolving climate conditions. Specifically, this study (i) provides a physically grounded estimate of present-day forcing for well-mixed greenhouse gases that does not require precise knowledge of the instantaneous atmospheric state, and (ii) introduces a computationally efficient framework that can be applied consistently to benchmark radiative forcing of all major well-mixed greenhouse gases across past and future climates, without requiring further massive line-by-line calculations. We believe this addresses a central and long-standing challenge in quantifying greenhouse gas forcing under realistic, all-sky conditions.

- *The manuscript introduces unnecessary confusion on a number of points. - The value of determining climate model-specific radiative forcing is that it helps disentangle variations in forcing from variations in response (feedbacks). The forcing to which each model is subject includes any errors in model-specific treatments of radiative transfer, so it’s unclear what is gained by “constraining” or “benchmarking” radiative forcing estimates from models contributing to CMIP6. (The models are in no sense an estimate of the uncertainty in forcing to which the earth is subject.) The roles of state dependence and model error in determining the diversity of model forcing are transparently available by applying off-line calculations (as is done here and in [20]); it’s not clear what value the use of a regression model adds.*

Error of model-specific radiative transfer parameterizations on LW IRF is directly diagnosed using the regression model. Limitations of offline calculations are discussed

in **L68–77** and **L130–133**, and the value added by the regression approach is further addressed as follows.

Offline line-by-line calculations are **computationally prohibitive** to apply comprehensively across all models and greenhouse gas forcing scenarios. For example, the line-by-line calculations used by He et al. (2023) are available for only two models (the same datasets submitted to RFMIP; Pincus et al., 2020) and cannot be practically extended to all CMIP-class models. While the offline SOCRATES calculations in He et al. (2023) are computationally efficient, they are not designed to serve as a global benchmark across models. In contrast, the regression framework introduced here can be readily applied to every model, every grid point, and every time instance, allowing systematic diagnosis of radiative parameterization biases for each major well-mixed greenhouse gas, with global-mean accuracy comparable to line-by-line calculations for benchmark purposes.

We also note that the offline calculations in He et al. (2023) are conducted under clear-sky conditions. In practice, **offline calculations cannot provide a definitive benchmark under all-sky conditions**, because subgrid-scale cloud–radiation interactions are nevertheless parameterized and highly model-dependent, and therefore no all-sky ground truth can be simulated for offline evaluation. The regression method naturally incorporates each model’s cloud–radiation interactions, because clouds do not alter the relationship between OLR and LW forcing, as examined in Methods and Extended Data Fig. 4.

- *The focus on instantaneous radiative forcing at the tropopause is not consistent with the last several decades of thinking. Interest in radiative forcing at the troposphere (e.g. Hansen et al. 1997, doi:10.1029/96JD03436) arose before stratospheric cooling (which really affects only CO₂, as [34] shows) was understood as one of many possible radiative adjustments. It’s interesting that radiative forcing is more linear in OLR at the tropopause than at the TOA but this does not make the former more relevant than the latter.*

As shown in Fig. 3, LW IRF at the tropopause explains most of the inter-model variability in effective radiative forcing, whereas TOA forcing shows a weaker relationship. This behavior is **consistent with longstanding understanding of radiative adjustments** (Fels, 1980; WMO, 1982; IPCC assessments over several decades, including Shine et al., 1990; Griggs et al., 2001; Collins et al., 2006; Myhre et al., 2013). Further discussion is provided in **L616–629**.

The reduced linearity between radiative forcing and OLR at the TOA has been fully addressed in Methods (**L583–629**). It primarily arises from seasonal variations in stratospheric temperature and lapse rate (L606–611).

- *The regression models used for predicting forcing from OLR depend on concentration in non-linear ways (Tables 1 and 2). Why should a linear relationship hold across more than W/m^2 in OLR but not across modest changes in the concentration of greenhouse gases?*

The linear relationship between LW IRF and OLR arises because variations in humidity and clouds induce proportional effects on both quantities, as demonstrated in Extended Data Fig. 4.

There is no physical reason for the dependence on gas concentration to be similar to the dependence on OLR. A quadratic term was introduced during revision solely to maintain consistent numerical accuracy over a wide range of perturbations in a unified regression framework for all gases, and to be numerically consistent with the direct output from our Code Ocean script (<https://doi.org/10.24433/CO.0711190.v2>). When concentrations are expressed in ppmv, the quadratic contribution is very small, typically on the order of 1% of the total. This clarification has been added in **L572–576** and the Table 1 footnote.

- *The manuscript conflicts with several existing results but does not mention or attempt to reconcile these conflicts. In particular: - Reference 20 reports that much of the variability in radiative forcing due to CO₂ arises from differences in stratospheric temperature; here the authors claim (lines 236–237) that the calculation of LW IRF is the dominant cause of spread.*

Extended Data Table 4 shows that LW IRF at the tropopause explains approximately 91% of the inter-model spread, whereas IRF at the TOA explains about 48% (see also **L226–232**). Our conclusion does not conflict with He et al. (2023), which emphasizes that stratospheric temperature is a major contributor to spread in IRF at the TOA.

- *Reference 27 reports that radiative forcing by halocarbons is strictly linear in concentration across a much wider range of concentrations than is explored here, but Tables 1 and 2 use quadratic forms.*

We agree that radiative forcing by HFCs is effectively linear in concentration. The quadratic terms used in Tables 1 and 2 are numerically very small when concentrations are expressed in ppmv (HFCs are in the order of 10^{-4} ppmv, when combined with $(C-C_0)^2$, it is naturally 10^{-4} smaller than the $(C-C_0)$ term) and do not contradict the results of Forster et al. (2005). The small contribution by the quadratic term can be numerically neglected depending on user’s application. This point is now clarified explicitly in **L572–576** and the Table 1 footnote.

Referee 2

- *The authors have satisfactorily addressed all the points raised in my first review. I have no further comments.*

-Florian Roemer

ETH Zurich

We thank the referee for the assessments and providing the addendum. The role and magnitude of the second-order terms have been clarified in the revised manuscript to avoid potential confusion.

Referee 3

- *I thank the authors for their extensive revisions.*

The revised manuscript is much improved from the original submission and satisfactorily addresses the points I raised and appears to address the issues raised by the other reviewers too. Furthermore, the importance and impact of the manuscript is now clear. It also provides a strong motivation for the scientific community to (continue to) measure OLR accurately to track GHG RF.

My one minor remaining comment is that given the findings of this manuscript, I would recommend that the authors reference the issues/concerns raised in Soden et al. (2018) and then comment in the manuscript's conclusion about how/whether their work addresses those issues or if there is still more work to be done.

We thank the referee for this insightful suggestion. We have now cited Soden et al. (2018) in the Introduction and Conclusion to connect our findings to the long-standing concerns it raises regarding uncertainties in CO₂ radiative forcing (**L320–322**).